



# Observation and modeling of high-[7]Be events in Northern Europe associated with the instability of the Arctic polar vortex in early 2003

Erika Brattich[1], Hongyu Liu[2], Bo Zhang[2], Miguel Ángel Hernández-Ceballos[3], Jussi Paatero[4], Darko Sarvan[5], Vladimir Djurdjevic[6], Laura Tositti[7], and Jelena Ajtić[5]

[1] Department of Physics and Astronomy DIFA, Alma Mater Studiorum University of Bologna, via Irnerio 46, 40126 Bologna (BO), Italy

[2] National Institute of Aerospace, 100 Exploration Way, Hampton, VA 23666, USA

[3] Department of Physics, University of Cordoba, Rabanales Campus, 14071 Cordoba, Spain

[4] Finnish Meteorological Institute, P.O. Box 503, FI-00101, Helsinki, Finland

[5] Faculty of Veterinary Medicine, University of Belgrade, Bulevar oslobođenja 18, 11000 Belgrade, Serbia

[6] Institute of Meteorology, Faculty of Physics, University of Belgrade, Studentski trg 18, 11000 Belgrade, Serbia

[7] Department of Chemistry "G. Ciamician", Alma Mater Studiorum University of Bologna, via Selmi 2, 40126 Bologna (BO), Italy

*Correspondence to*: Erika Brattich (erika.brattich@unibo.it)

**Abstract.** Events of very high concentrations of [7]Be cosmogenic radionuclide have been recorded in the subpolar regions of Europe during the cold season. With an aim to investigate the mechanisms responsible for those peak [7]Be events, and in particular to verify if they are associated with the fast descent of stratospheric air masses occurring during sudden stratospheric warming (SSWs), we analyse [7]Be observations at six sampling sites in Scandinavia during January-March 2003 when very high [7]Be concentrations were observed and the Arctic vortex

was relatively unstable as a consequence of several SSWs. We use the GEOS-Chem chemistry and transport model driven by the MERRA-2 meteorological reanalysis to simulate tropospheric [7]Be over Northern Europe. We show that the model reasonably reproduces the temporal evolution of surface [7]Be concentrations observed at the six sampling sites. Our analysis of model simulations, surface [7]Be observations, as well as atmospheric soundings of ozone and temperature indicates that the [7]Be peak observed in late February 2003 (between 20 and 28 February

2003) at the six sampling sites in Scandinavia was associated with downward transport of stratospheric vortex air originated during SSW that occurred a few days before the peak (18-21 February 2003).

## 1 Introduction

Beryllium-7 ([7]Be) is a cosmogenic radionuclide widely monitored and analysed around the world (e.g.,

Tositti et al., 2004, 2014; Gourdin et al., 2014; Sýkora et al., 2017). Due to its relatively long radiative half-life





(53.22 days) and its cosmogenic origin in the upper troposphere and lower stratosphere (UT-LS) (Lal and Peters, 1967), [7]Be is considered a tracer of stratospheric influence and large-scale subsidence (e.g., Liu et al., 2016; Chae and Kim, 2019). The variability of the [7]Be activity concentration in surface layers is driven by both static and dynamic factors, e.g., geographical location of the monitoring sites (e.g., Hernández-Ceballos et al., 2015), seasonal

atmospheric processes driving transport of carrier aerosols (Lal and Peters, 1967), stratosphere-troposphere air mass exchange (Cristofanelli et al., 2003; Cristofanelli et al., 2009; Putero et al., 2016; Brattich et al., 2017a), synoptic and mesoscale patterns, vertical transport in the troposphere (Lee et al., 2007), solar activity and dry and wet deposition (e.g., Hernández-Ceballos et al., 2015, 2016a; Ioannidou and Papastefanou, 2006).

The spatial and temporal variability of the [7]Be surface concentrations in Europe and their relationship with
meteorological variables was previously analysed in many studies (e.g., Piñero García et al., 2012; Błażej and Mietelski, 2014). The impact of the 11-year solar modulation on the [7]Be concentrations in the air is well established (e.g., Leppänen et al., 2010). The distinctive spring/summer maximum of [7]Be concentrations is widely described and mainly linked with the increased downward transport from the upper troposphere resulting from the intense convection and higher tropopause height typical of the warm season (Cristofanelli et al., 2006; Gerasopoulos et al.,
2001, 2003). In addition, cases of high [7]Be surface concentration, some of which occurring over the autumn/winter season, have been analysed in Europe, e.g., over the Iberian Peninsula (Hernández-Ceballos et al., 2017) and at high-altitude stations in the Alps and the Apennines (Brattich et al., 2017; Cristofanelli et al., 2006, 2009). The spring/summer maximum was originally observed with fission products injected into the stratosphere during atmospheric nuclear tests (Dutkiewicz and Husain, 1985; Cristofanelli et al., 2018). A study of Salminen-Paatero
et al. (2019) using potential vorticity analysis indicated that the transfer of stratospheric air into the troposphere was at its maximum in March followed by gradual movement into the ground-level air during spring and early summer.

Further, over the last decade many studies have investigated the [7]Be records in Northern Europe (Leppänen et al., 2010, 2012; Leppänen and Paatero, 2013; Sarvan et al., 2017; Leppänen, 2019), one of the three regions in
Europe identified with a distinct [7]Be behaviour in the surface air (Ajtić et al., 2017; Hernández-Ceballos et al. 2015, 2016b). Among these studies, Ajtić et al. (2016) analysed the [7]Be concentration measured in Helsinki, Finland, over 25 years (1987-2011), and pointed out a relatively high number of [7]Be extremes occurring over autumn and winter: more specifically, 10 % of the highest [7]Be concentrations (above 95[th] percentile) were observed in the cold season (October-March). Furthermore, recent studies have also indicated that the polar vortices can have a notable
influence on the wintertime [7]Be surface concentrations in both the Northern (Ajtić et al., 2018; Bianchi et al., 2019; Terzi and Kalinowski, 2017) and Southern Hemispheres (Pacini et al., 2015).



In particular, Ajtić et al. (2018) and Bianchi et al. (2019) employed two different methodologies to identify episodes of extremely high $^7$Be surface concentrations in the autumn and winter seasons, pointing out a large number of cases over the October-March period of the years investigated. The comparison of the dates identified

in both analyses showed an overlap with the events of the so-called Sudden Stratospheric Warming (SSW) of the Arctic vortex, i.e., a sudden rise in the polar temperatures that leads to a highly irregular shape of the vortex and its misalignment from the pole. Ajtić et al. (2018) also noted cases of extremely high $^7$Be concentrations occurring right after a very low $^7$Be concentration over the Scandinavian Peninsula during autumn and winter. Overall, this relationship between the SSW of the Artic vortex and high $^7$Be surface concentrations is likely linked to the

perturbed stratosphere-troposphere interactions associated with SSWs, which could favor a fast descent of: 1) midlatitude air rich in $^7$Be, thus increasing this radionuclide's surface abundance, and 2) aged vortex air wherein $^7$Be is subjected to radioactive decay and not transported from outside the vortex, thus decreasing the $^7$Be surface abundance.

The atmospheric circulation in the Arctic is dominated by the presence of two distinct polar vortices, one

in the troposphere and one in the stratosphere (Waugh et al., 2017). The two vortices present well distinct features: first, the vortex in the troposphere is much more extended than the stratospheric one; and second, while the tropospheric vortex is present all-year round, the stratospheric polar vortex exists only from fall to spring (Waugh et al., 2017). The stratospheric polar vortex in winter stems from the large-scale temperature gradients between the midlatitudes and the poles. Therefore, the stratospheric polar vortex begins to form in autumn as a result of the

decreasing solar heating in the polar regions; it strengthens during winter and then breaks down in spring when solar radiation returns to the polar region. Larger topographic and land-sea contrasts and the resulting stronger upward-propagating waves in the Northern Hemisphere, make the northern stratospheric vortex, or the Arctic vortex, weaker and more distorted than its Southern Hemisphere counterpart, the Antarctic vortex. A higher temporal variability of the Arctic vortex includes the SSW. A major SSW can even cause the stratospheric vortex

to break down during midwinter (Waugh et al., 2017).

While the initial scientific interest over the stratospheric polar vortex was especially linked to the stratospheric ozone depletion and formation of the ozone holes over the poles, it is now recognized that the vortices might affect the processes in the troposphere and surface weather (e.g., Mitchell et al., 2013). The present work aims at investigating in detail the atmospheric processes responsible for high $^7$Be activities recorded in the cold

season over the Scandinavian Peninsula and its relationship with the Arctic polar vortex through model simulations. For this purpose, we conduct $^7$Be simulations for the period of January-March 2003 using the GEOS-Chem global 3-D chemical and transport (CTM) model. The period was selected because of the large number of events with





extremely high $^7$Be concentrations at surface in Scandinavia; some of these events were preceded by very low surface concentrations in the lower troposphere ($< 10^{th}$ percentile). This period thus offers the opportunity to test

the hypothesis that SSWs ease a fast descent of not only the midlatitude but also vortex air (Ajtić et al., 2018.). To achieve this goal, our analysis will therefore focus on:

- Investigating the processes responsible for the variability of $^7$Be concentrations in surface air in Northern Europe;

- Better understanding whether and how SSW and the Arctic vortex winter-time instability influence the
surface concentrations of $^7$Be in Northern Europe;

- Quantifying the rate of air subsidence on the inner and outer side of the vortex during the period of its instability.

To analyse the influence of SSW and of the Arctic polar vortex on $^7$Be concentrations, we first assess the performance of the GEOS-Chem model in reproducing the observed $^7$Be variability. We then use model simulations

together with other supporting measurements from soundings and meteorological datasets to examine the processes responsible for the variability in the $^7$Be concentrations over the period of January-March 2003.

As opposite to the cosmogenic origin of $^7$Be, $^{210}$Pb (half-life 22.3 years) is a nuclide of crustal origin derived from decay of $^{222}$Rn (half-life 3.8 days), which is emitted from soils by decay of $^{226}$Ra. Owing to the contrasting natural origins of the two nuclides, the $^7$Be/$^{210}$Pb ratio is often regarded as indicative of vertical transport processes

and convective activity in the atmosphere (e.g., Koch et al., 1996; Tositti et al., 2004; Brattich et al., 2017a,b). After being produced by contrasting physical mechanisms, both $^7$Be and $^{210}$Pb rapidly attach to ambient submicron-sized particles (e.g., Gaffney et al., 2004) and are removed by wet (mainly) and dry (secondarily) deposition processes of their carrier aerosol. The bias in the simulated $^7$Be/$^{210}$Pb ratio due to uncertainties in the model deposition schemes is thus reduced. For this reason, besides $^7$Be, the $^7$Be/$^{210}$Pb ratio was also analysed to gain

further insights into vertical transport processes during the study period.

The rest of this paper is organized as follows. Section 2 describes the radioactivity ($^7$Be and $^{210}$Pb) and meteorological data used. Section 3 provides a brief description of GEOS-Chem, the HYSPLIT trajectory model, and statistical parameters used to assess the model's performance in reproducing the observations. Section 4 presents an overview of the $^7$Be observations made in Northern Europe in the boreal winter 2003. Section 5 analyses

the precipitation and transport pattern in the study region, while Section 6 assesses how the GEOS-Chem model performs in reproducing the observed variability in the monthly mean surface $^7$Be concentrations during the study period. Section 7 further evaluates the performance of the model in reproducing the short-time variability of $^7$Be in



Northern Europe, followed by interpreting the observed variability using model simulations and additional meteorological observations in section 8. Finally, summary and conclusions are given in Section 9.

## 2 Data

In this section we briefly describe the $^7$Be and $^{210}$Pb radioactivity data as well as the meteorological datasets analysed in this work.

### 2.1 $^7$Be data

Since 1988, the Radioactivity Environmental Monitoring data bank (REMdb) (https://rem.jrc.ec.europa.eu/RemWeb/) has brought together and stored in a harmonised way environmental radioactivity data (air, water, milk and mixed diet) measured by the European Member States (Sangiorgi et al., 2019). Among the set of sample types and measurements recommended in 2000/473/Euratom (European Commission, 2000), measurements of natural radioelements, such as $^7$Be in surface air, are required, and hence, it is very closely monitored and widely stored in REMdb (De Cort et al., 2007).

Within the REMdb, the activity concentration of $^7$Be in the surface air in Northern Europe (latitude north of 55°N) is available for six sampling sites (Hernández-Ceballos et al., 2015): Ivalo, Umea, Helsinki, Kista, Harku and Risoe, however, with varying start dates and sampling frequencies (Figure 1a). The largest dataset is for Helsinki where, since 1999, the sampling has been performed daily or once every two days. Datasets for Ivalo, Umea, Kista and Risoe also span more than two decades and have a good temporal coverage (roughly once a week since 1995).

### 2.2 $^{210}$Pb data

Daily aerosol samples were collected in Helsinki on the roof of the Finnish Meteorological Institute's main building (60°10'N, 24°57'E). Filters (Munktell MGA, diameter Ø = 240 mm) were changed every day at 06 UTC. The air volume was about 3500 m³/day. The filters were assayed for $^{210}$Pb by alpha counting of the in-grown daughter nuclide $^{210}$Po (Mattsson et al., 1996).

### 2.3 Meteorological data

The $^7$Be variability is tightly linked to horizontal and vertical transport of the carrier aerosol, and to precipitation that leads to the radionuclide's removal from the atmosphere. Here we use the Modern-Era





Retrospective analysis for Research and Applications, Version 2 (MERRA-2) meteorological reanalysis (Gelaro et

al., 2017) to assist in the data analysis and to drive the GEOS-Chem model simulations. MERRA-2 is produced with version 5.12.4 of the Goddard Earth Observing System (GEOS) atmospheric data assimilation system. It assimilates modern observations of the atmosphere, ocean, land, and chemistry, and includes assimilation of aerosol remote sensing data.

        Vertical soundings of air temperature from the Finnish Meteorological Institute's (FMI) Arctic Space

Centre (http://fmiarc.fmi.fi) at Sodankylä, northern Finland (67.37°N, 26.63°E) were obtained from the University of Wyoming (http://weather.uwyo.edu/upperair/sounding.html). Ozone sounding data (Kivi et al., 2007; Denton et al., 2019) was retrieved from the database of the FMI's Arctic Space Centre (http://litdb.fmi.fi). To study the effect of downward transport of stratospheric air masses into the troposphere, potential vorticity (PV) values (Holton et al., 1995) were calculated from wind, temperature, and surface pressure fields obtained from the European Centre

for Medium-Range Weather Forecasts (ECMWF), Reading, UK.

## 3 Methods

        In this section we give a brief description of the GEOS-Chem and HYSPLIT models and the statistical parameters used to indicate the model performances.

### 3.1 GEOS-Chem model

GEOS-Chem (http://www.geos-chem.org) is a global 3-D CTM that has been widely used to study atmospheric composition and processes (e.g., Bey et al., 2001; Park et al., 2004; Eastham et al., 2014). In this study, we use the GEOS-Chem v11-01f to simulate $^7$Be and $^{210}$Pb and assist in interpreting the observations. GEOS-Chem includes a radionuclide simulation option ($^{222}$Rn-$^{210}$Pb-$^7$Be), which simulates the emission, transport (advection, convection, boundary layer mixing), deposition and decay of the radionuclide tracers (Jacob et al., 1997; Liu et al.,

2001; Yu et al., 2018).

        We use the $^7$Be production rates recommended by Lal and Peters (1967) for a maximum solar activity year (1958), which has been shown to produce the best results compared to aircraft $^7$Be observations (Koch et al., 1996; Liu et al., 2001). The production rate is formulated as a function of latitude and pressure without seasonal variation (Koch et al., 1996). About two thirds of atmospheric $^7$Be is generated in the stratosphere. $^{222}$Rn emission follows a

recent work by Zhang et al. (2020), in which a customized emission map was built upon a few previously published emission scenarios and evaluated against global $^{222}$Rn surface observations and aircraft profiles. $^{222}$Rn emission





flux rate is a function of latitude, longitude, and month. [7]Be and [210]Pb are assumed to behave like aerosols once formed in the atmosphere and subject to dry and wet deposition (Liu et al., 2001). Both wet and dry deposition for [222]Rn are neglected due to its inert nature.

GEOS-Chem simulations in this work are driven by the MERRA-2 meteorological reanalysis. The native resolution of MERRA-2 is 0.667° longitude by 0.5° latitude, with 72 vertical layers (top at 0.01hPa). The meteorological fields are regridded into 2.5° longitude by 2° latitude for the GEOS-Chem simulations in this work. GEOS-Chem uses the TPCORE advection algorithm of Lin and Rood (1996). Convective transport is calculated using archived convective mass fluxes (Wu et al., 2007). Boundary-layer mixing is based on the non-local scheme

implemented by Lin and McElroy (2010). The wet deposition scheme follows that of Liu et al. (2001) and includes rainout (in-cloud scavenging) due to stratiform and anvil precipitation, scavenging in convective updrafts (Mari et al., 2000), and washout (below-cloud scavenging) by precipitation (Wang et al., 2011). Precipitation formation and evaporation fields are archived in MERRA-2 and used directly by the model wet deposition scheme.  Dry deposition is based on the resistance-in-series scheme of Wesely (1989).

In addition to the standard model simulations of [7]Be and [210]Pb, we also separately transport [7]Be produced in the stratosphere to quantify the stratospheric contribution to [7]Be in the troposphere. All model simulations are conducted for the period of January 2002 – March 2003 with initial conditions from previously archived restart files. Hourly and monthly mean outputs for January-March 2003 are used for analysis.

### 3.2 HYSPLIT

The Hybrid Single Particle Lagrangian Integrated Trajectory (HYSPLIT) model, developed by the NOAA's Air Resources Laboratory (ARL) (Stein et al., 2015), was used to calculate a set of backward trajectories during the study period. To compute the 96 h 3D backward trajectories at 00, 06, 12, and 18 UTC and with different ending heights: 100, 500, 1000 and 1500 m above ground level, the NCEP (National Centers for Environmental Prediction) FNL Operational Global Analysis (NCEP/NWS/NOAA/U.S. Department of Commerce, 2000) meteorological files

were used. While 96 h was considered a sufficiently long period to represent the synoptic air flows, the heights were selected to help us to understand the behaviour of the airflows circulating in the Atmospheric Boundary Layer (ABL), just above the ABL, and in the free troposphere. We used the cluster methodology implemented in the HYSPLIT model to group the calculated trajectories according to their length and curvature, and thus identify the airflow patterns over the whole period of the analysis. It is worth mentioning that clusters, as well as trajectories,

indicate an estimation of the general airflow rather than the exact pathway of an air parcel (e.g., Jorba et al., 2004; Salvador et al., 2008).





**3.3 Evaluation of the model output**

The performance of the model in reproducing observed activity concentrations is evaluated by calculating some basic statistical parameters, such as the mean and standard deviation and other indicators, according to the 210 methodology developed by Hanna (1993) and summarized later by Chang and Hanna (2004). Specifically, the performance of the CTM was evaluated using the following set of indicators, proposed by Carruthers et al. (2004):

- The mean bias (*MB*), a measure of the mean difference between the modelled and observed concentrations:

$$MB = \overline{C_m - C_o} \qquad (1)$$

where $C_m$ is modelled concentration and $C_o$ is observed concentration.


- The normalized mean square error (*NMSE*), a measure of the mean difference between matched pairs of modelled and observed concentrations:

$$NMSE = \frac{\overline{(C_m - C_o)^2}}{\overline{C_m}\,\overline{C_o}} \qquad (2)$$

- The fraction of modelled concentrations within a factor of 2 of observations (*FA2*), i.e., for which $0.5 <$ 220 $C_m/C_o < 2$

- The Pearson's correlation coefficient (*R*), a measure of the extent of a linear relationship between the modelled and observed concentrations:

$$R = \frac{\sum_{i=1}^{n}(C_{o.i} - \overline{C_o})(C_{m.i} - \overline{C_m})}{\sqrt{\sum_{i=1}^{n}(C_{o,i} - \overline{C_o})^2\,\sum_{i=1}^{n}(C_{m,i} - \overline{C_m})^2}} \qquad (3)$$

A perfect model has MB and NMSE values equal to 0 and FA2 value equal to 1, while the R results range 225 from −1 (perfect negative relationship) to + 1 (perfect positive relationship), where 0 implies no relationship between the variables. To better understand the quantitative differences between observations and simulations, scatter plots were used.

**4 Boreal winter 2002/2003**

As indicated by Ajtić et al. (2018) and Bianchi et al. (2019), the winter of 2003 offers a good opportunity 230 to investigate a possible link between SSWs and extreme surface concentrations of [7]Be detected in Northern Europe. This period is sufficiently covered by the [7]Be activity concentration measurements at all six monitoring sites.





In particular, very high [7]Be activity concentrations, above the 90[th] percentile simultaneously at most of the Scandinavian Peninsula sampling sites, were recorded around 23-24 February 2003 (Ajtić et al., 2016, 2018)

(Figure 1b). During the 2002/2003 boreal winter, the Arctic vortex was relatively unstable, with six SSWs taking place over the whole season (Peters et al., 2010). Two very pronounced episodes, which were both associated with the vortex splitting and fast SSW recovery, occurred in January and February, respectively (Günther et al., 2008). The evolution of the vortex caused vortex filamentation and vigorous mixing of the vortex and midlatitude stratospheric air (Günther et al., 2008; Müller et al., 2003). Several balloon flights inside the Arctic polar vortex in

early 2003 observed unusual trace gas distributions connected to an intrusion of mesospheric air down to altitudes of about 25 km (Engel et al., 2006; Huret et al., 2006; Müller et al., 2007). Since such disturbances around the pole are expected to affect the troposphere, i.e., on weather conditions (Baldwin and Dunkerton, 2001), and air composition (Hsu, 1980; Limpasuvan et al., 2004). Hence, the high [7]Be concentrations that were measured in Scandinavia around 24 February 2003 could be a result of downward motion of midlatitude stratospheric air.

Interestingly, prior to this episode, very low (below the 10[th] percentile for each site) surface concentrations of [7]Be were measured in Risoe, Kista and Ivalo on 3, 10 and 16 February 2003, respectively (Figure 1). These low values were tentatively linked by Ajtic et al. (2018) with the transport of aged stratospheric vortex air poor in [7]Be, even though they are more likely related to precipitation scavenging that occurred the days before, as shown by the ECA&D (European Climate Assessment & Dataset, https://www.ecad.eu/) records. The reader is referred to Ajtić

et al. (2018) for more details.

## 5 Analysis of winter precipitation and transport in the Scandinavian Peninsula: observations vs. model simulations

Before analysing the temporal pattern of simulated [7]Be concentrations, we analysed the precipitation and transport pattern in the MERRA-2 meteorological dataset that drives the GEOS-Chem simulations. In particular,

the MERRA-2 precipitation was evaluated against the data from Global Precipitation Climatology Project (GPCP) satellite (ftp://meso.gsfc.nasa.gov/pub/gpcp-v2.2/psg) and surface observations in winter 2003.

Figure 2 shows the MERRA-2 and GPCP monthly precipitation in winter for the region within 0-90°N and 90°W – 90°E. Good agreement is found between the MERRA-2 and the GPCP precipitations averaged over the region. Specifically, the geographical distribution of precipitation in MERRA-2 shows some important features that are consistent with the observed climatology precipitations: the desert climate in North Africa with very low





precipitation throughout the period, high precipitation over the North Atlantic region during winter, and Europe where the seasonal pattern of precipitation is similar to that in the North Atlantic region.

To assess the capability of the model to correctly capture the trend in precipitation during the observation period at the sampling sites, we examined the normalized differences between the MERRA-2 and the observed precipitation, calculated as a difference between the MERRA-2 and the observed values, normalized over the observed value (Table 1).

Overall, the MERRA-2 precipitation tends to be generally higher than that of GPCP at all sampling sites (Table 1) except for Harku and Helsinki, and especially in the February-March period. This result is in agreement with the findings of Gelaro et al. (2017) who compared the global precipitation of MERRA-2 and GPCP, and reported a general positive bias over northern high latitudes. However, the agreement between MERRA-2 and GPCP precipitation seasonality is reasonable, as indicated by the correlation coefficient values, higher than 0.85 at all sites except for Ivalo (-0.32), and the low NMSE values, in the range of 0-0.42 (Table 1).

Figure 3 shows that winter circulation in the Scandinavian Peninsula is dominated by SW and W winds (Chen, 2000; Linderson, 2001). The analysis of the main circulation in the three months in Figure 3 reveals low wind speeds from S-SW in the study area and period. A region of strong wind speeds, possibly corresponding to the Arctic vortex, is clearly visible at surface level to the west of the study area in all the three months. In addition, there appears to be a convergence area (opposite wind directions) between 60 and 75°N. Model-simulated $^7$Be/$^{210}$Pb ratios and fraction of stratospheric $^7$Be increase over the three months period and peak in March, suggesting increasing stratospheric influence, subsidence, or convective mixing in the study region.

## 6 Variations of the monthly mean surface $^7$Be concentrations in the Arctic region: model simulations vs. observations

Figure 4a shows a scatter plot comparing the simulated and observed monthly mean $^7$Be concentrations at the six sampling sites. Table 2 reports the statistical parameters and the normalized differences that indicate the performance of the GEOS-Chem model in reproducing the observed $^7$Be monthly means.

In general, the model well simulates the month-to-month trend in $^7$Be concentrations measured at the sampling sites, as indicated by the fact that all the values fall within the 95% confidence levels (Figure 4a) and the high positive correlation coefficients (> 0.7) except for Ivalo and the low MB and NMSE values (Table 2). In fact, the normalized differences are not very high (generally <1), except at Risoe. The bias between the model and the observations is partly attributed to the coarse resolution of the model. Overall, the simulations underestimate the





290    observed values, likely due to uncertainties associated with the deposition schemes and/or precipitation as discussed earlier.

The use of the $^7$Be production rate of Lal and Peters (1967) for a solar maximum year (1958) may also partly explain the tendency of simulated $^7$Be to be lower than observed. The sunspot number in 2003 (99.3) was rather low (slowly decreasing from 2000, a solar maximum year, and reaching minimum in 2008) if compared to 295    the 1958 value of 184.8. As known, the galactic cosmic-ray intensity, largely responsible for the production of cosmogenic radionuclides, at the Earth's orbit is inversely related to solar activity (O'Brien, 1979), leading to the well-known phase opposition between sunspot number and $^7$Be concentration (e.g., Hernández-Ceballos et al., 2015). Sunspot number data herein used were extracted from the World Data Center for the production, preservation and dissemination of the international sunspot number (Sunspot Index and Long-term Solar 300    Observation, SILSO, Royal Observatory of Belgium, Brussels, http://www.sidc.be/silso/datafiles - total).

## 7 Variations of the $^7$Be weekly and daily mean surface concentrations in the Arctic region: observations vs. model simulations

After analysing the model's performance in reproducing $^7$Be monthly mean observations in the previous Section, we compare here in Figure 4b simulated and observed weekly (daily in the case of Helsinki) $^7$Be activity 305    concentrations at the six sampling sites. Table 3 shows the corresponding parameters that indicate the performance of the GEOS-Chem model in reproducing observations. The weekly evolution of simulated versus observed $^7$Be concentrations at these sites is shown in Figure 5.

As with the monthly means, the model generally represents adequately the temporal pattern but not the magnitude of weekly mean concentrations, which tend to be lower than those observed (Table 3, Figures 4b and 310    5). This can arise from the higher precipitation in the model than in the observations and/or from errors in the deposition schemes. However, the correct reproduction of the $^7$Be temporal pattern, as indicated by the high correlation values at all sampling sites with the exception of Risoe, suggests that the model captures the transport processes leading to the peak in $^7$Be concentrations at the end of February 2003 and the preceding very low concentration values. In addition, the low MB and NMSE values calculated at all sites and especially at Ivalo and 315    Umea suggest that the model reproduces adequately the observed values (Figure 5).

As for the $^7$Be/$^{210}$Pb ratio in Helsinki (Table 3), the model tends to underestimate the observed ratio, which could be due to the model underestimating $^7$Be and overestimating $^{210}$Pb. Nevertheless, the relatively high





correlation between the simulated and observed ratios suggests a reasonable simulation of the temporal pattern of this tracer.

### 8 Understanding the $^7$Be variations during the 2002/2003 boreal winter


As mentioned earlier (Section 4), an SSW event occurred at the end of February 2003. We concentrated our analysis on two different periods during the month: early in the month, between 3 and 16 February when very low $^7$Be concentration values were recorded, and at the end of the month between 20 and 28 February characterized by extremely high $^7$Be concentrations. To gain further insights into the $^7$Be variations during the 2002/2003 boreal

winter (Section 7), we analysed the simulated $^7$Be/$^{210}$Pb ratio, maps of surface winds and relative humidity, ozone soundings, vertical cross sections of simulated $^7$Be activity concentrations and calculated potential vorticity, and simulated and observed vertical profiles of air temperature. The results were further supported with the analysis of the clusters of back-trajectories during the two different periods of low and high $^7$Be concentrations.

During the 2002/2003 boreal winter, the Arctic vortex was relatively unstable, with six SSWs taking place

over the whole season (Peters et al., 2010), and two very pronounced SSWs, with the vortex splitting and re-establishing within a few days, occurred in January and February (Günther et al., 2008). On one hand, this evolution of the vortex caused vortex filamentation and vigorous mixing of the vortex and midlatitude stratospheric air (Günther et al., 2008; Müller et al., 2003). On the other hand, such disturbances around the pole were also expected to propagate through the troposphere, i.e., on weather conditions (Baldwin and Dunkerton, 2001), and air

composition (Hsu, 1980; Limpasuvan et al., 2004). Hence, the high $^7$Be concentrations measured in Scandinavia around 24 February 2003 could be the result of downward transport of midlatitude stratospheric air. Interestingly, prior to this episode, very low (below the 10[th] percentile for each site) surface concentrations of $^7$Be were measured in Risoe, Kista and Ivalo on 3, 10 and 16 February 2003, respectively (Figure 1b), as connected to precipitation scavenging that occurred the days before, as discussed earlier.


Figure 6 presents the temporal (weekly mean) pattern of $^7$Be/$^{210}$Pb and of the stratospheric fraction of $^7$Be (calculated as the ratio of the stratospheric $^7$Be tracer concentration to the total $^7$Be concentration in the troposphere) at the six sampling sites, while daily observations of the $^7$Be/$^{210}$Pb ratio at Helsinki and Sodankylä (67.367°N, 26.629°E; 160 km south of Ivalo) are presented in the Supplementary Information (hereafter SI). At the beginning of February, the ratio was generally quite low at all the sites. In contrast, the week of 19-26 February 2003 was

marked by an evident peak in the $^7$Be/$^{210}$Pb ratio and a simultaneous increase in the fraction of $^7$Be originating in





the stratosphere at all sites, which together could be the first indication of a prominent vertical transport from the UT-LS region.

Hence, we further examined the vertical profiles of temperature with an aim to identify differences in vertical transport near the beginning and end of February. The soundings from the Sodankylä station in the Arctic

offer three sets of measurements for each of the investigated periods: on 10 and 16 February, that fall into the period when very low [7]Be concentrations were recorded in Kista and Ivalo, respectively; and 22 and 24 February, the days marked by extremely high [7]Be concentrations over the Scandinavian Peninsula; 20 and 21 February in the period of transition to high [7]Be concentration over the Scandinavian Peninsula. Figure 7 shows air temperature profiles in the MERRA-2 dataset and atmospheric soundings at the Sodankylä station. Besides the very good agreement

between the simulated and observed temperatures, a warming of the stratosphere (20-60 km) and a different vertical temperature structure of the lower stratosphere around 20-24 February as compared to the 10 and 16 February profiles are also evident. This observation is a clear indication of the link between the SSW and the [7]Be peak observed at the six sampling sites located in Northern Europe. In addition, the ozone soundings at the Sodankylä station reveal an ozone mixing ratio peak in the lower troposphere (~1.5-3km) on 19[th] February 2003 as compared

to those observed during 12, 26 and 28 February 2003 (Figure 8), consistent with downward transport from higher altitudes around that day. Despite the chemical ozone loss in the Arctic vortex in the stratosphere in 2003 as observed by ozone soundings (Tilmes et al., 2006), obviously lower-stratospheric ozone was still enhanced relative to tropospheric ozone.

Simultaneously, the analysis of maps of surface transport and relative humidity (Figure 9) highlights the

different winds and relative humidity values in the two periods, with low relative humidity values (~40-50%) suggesting subsidence from 18 to 21 February, and the transition from a clockwise circulation to the fast and complex wind system typical of the second period corresponding to the high [7]Be peak and the SSW. Interestingly, the parcel of the lowest relative humidity values occurs during the 18-21 February period, i.e., a couple of days before the dates of the [7]Be peaks in the measurements and those peaks in the simulated [7]Be/[210]Pb ratio and

stratospheric [7]Be fraction (Figure 1b and Figure 6). Together with enhanced ozone concentrations observed in the lower troposphere on 19[th] February, this suggests that the downward transport from the UT-LS was triggered by the SSW occurring a few days before.

To better constrain the stratospheric origin of the air masses arriving at the sampling sites during the two periods, we further analyzed the potential vorticity data from ECMWF during the month of February 2003 at three

latitudes (63, 64.5 and 66°N) along the 21°E meridian (Figure 10). The data reveal clearly a bubble of high potential vorticity down to the surface at the three latitudes from 18 till 22 February 2003, particularly at the northernmost

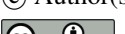



latitude where values are higher than 1.6 PVU, a value considered as a threshold for stratospheric air in the lower troposphere especially when in conjunction with low relative humidity, high $^7$Be/$^{210}$Pb ratios and ozone (Cristofanelli et al., 2006) were observed.

The low relative humidity (Figure 9) and high potential vorticity (Figure 10) corresponded to high $^7$Be descending to lower atmospheric levels, as simulated by the model (Figure 11). The descending vertical motion from the upper vertical levels during the period is clearly visible in the MERRA-2 pressure vertical velocity (omega) fields sampled at the six sampling sites for the month of February 2003, especially for the northernmost sites (Figure 12). Further evidence of this is seen from the maps of pressure vertical velocity (Figure 13) and of the
stratospheric fraction of $^7$Be originated in the stratosphere (Figure 14) at 940 hPa in the boundary layer.

        Analysis and comparison of 4-day back-trajectories at each sampling station allowed a reconstruction of two distinct atmospheric circulation patterns in the two periods. Figure 15 shows the clustering results for three sampling sites (Ivalo, Harku, Risoe) during both periods. The stations are ordered as a function of decreasing latitude from high (upper panels) to low (lower panels). Only results at 1000 m are shown and cluster results for
other altitudes in the lower troposphere are similar. While the first period (with low $^7$Be values) presents the dominance of westerly winds (air masses flowing eastward), as typical of these latitudes (Zanis et al., 1999), the second period is characterized by a clockwise displacement of airflows with origins at higher altitudes (Figure 15). This pattern in the second period is better established at lower latitude stations (Helsinki, Harku, Kista and Risoe) compared to higher latitude ones (Ivalo, Umea). It likely results from the aged vortex (Günther et al., 2008) and
the SSW at the end of February, corresponding to a decrease in the MERRA-2 daily average height of the thermal tropopause on 21$^{st}$-22$^{nd}$ February at the Sodankylä station in Finland (Figure S3) during the SSW (Peethani et al., 2014; Wargan and Coy, 2016). Associated with these processes is the downward transport of stratospheric air previously identified with an independent approach.

## 9 Summary and conclusions

We have used a global 3-D model (GEOS-Chem) driven by the MERRA-2 meteorological reanalysis to simulate atmospheric concentrations of $^7$Be of cosmogenic origin for the period of January-March 2003. The aim was to verify the mechanisms responsible for the surface $^7$Be variabilities in Northern Europe, and to test the hypothesis that SSWs may facilitate fast descent of UT-LS vortex air to the surface. The period was selected as it involves two intense SSWs and observations of extremely high $^7$Be concentrations at six sampling sites in
Scandinavia.





Before using the model's output to investigate the processes responsible for $^7$Be variability in Northern Europe over the period, we evaluated the MERRA-2 precipitation fields against the GPCP satellite and surface observations. A generally good agreement was found both at regional scale and at the six sampling sites. Analysis of the wind fields in the study period indicates low wind speeds from S-SW in agreement with the major circulation patterns over the Scandinavian Peninsula in winter, and the presence of a region of strong wind speeds to the west of the study area, likely in connection with the Arctic polar vortex.

The model reproduces efficiently the $^7$Be and $^7$Be/$^{210}$Pb temporal (i.e., monthly and weekly) patterns at the six sampling sites in the study period, even though it tends to underestimate the observed surface $^7$Be concentrations. The lower modelled values are likely due to its coarse resolution (2.5° longitude by 2° latitude), lack of year-to-year variation in $^7$Be production rates, and uncertainties associated with precipitation scavenging.

In order to investigate the processes responsible for $^7$Be variability at the six sampling sites during the study period, and in particular to test whether the peak $^7$Be concentrations measured in Scandinavia around 24 February 2003 originated from fast descent of stratospheric vortex air facilitated by SSW, we analysed time-height cross sections of simulated $^7$Be and potential vorticity, vertical profiles of air temperature, maps of surface winds and relative humidity, and ozone soundings. The analysis of the temporal variations of simulated $^7$Be/$^{210}$Pb ratio and fraction of $^7$Be originated in the stratosphere indicates a peak during the week of 19-26 February 2003, suggesting downward transport from the UT-LS region. The latter is corroborated by a layer of ozone mixing ratio enhancements in the lower troposphere recorded by the soundings at the Sodankylä station on 19$^{th}$ February. Furthermore, the vertical profiles of air temperature indicate a warming of the stratosphere and a change in shape in to the vicinity of the tropopause region during the period, suggesting the link between the downward transport of the vortex air and SSW.

Our analysis of time-height cross sections of simulated $^7$Be concentrations, calculated potential vorticity and MERRA-2 pressure vertical velocity (omega) reveals the vertical downward transport to the surface a stratospheric air parcel characterized by high potential vorticity, high vertical velocity (in particular at Ivalo on 19$^{th}$ February) and high $^7$Be concentrations, further supporting the stratospheric origin of the air masses during the investigated period.

Additionally, low relative humidity and a change in the circulation pattern from slow, clockwise to fast, swirling winds occurred over the study area. The change in the circulation pattern and the downward transport of stratospheric air was verified by the analysis of the clusters of back trajectories during the periods of low and high $^7$Be concentrations, which showed a change from westerlies to airflows from upper vertical levels.

Altogether, these analyses confirm the link between the SSW and transport of stratospheric air to the surface, resulting in high surface $^7$Be concentrations observed in February 2003 in Scandinavia. Since more frequent SSWs are expected in a warmer climate (Kang and Tziperman, 2017), this link has important implications for the impact of climate change on atmospheric transport, tropospheric composition, and air quality in northern high-latitude regions.

**Data availability**

The $^7$Be and $^{210}$Pb observational data are described in Section 2.1 and 2.2, respectively. The $^{222}$Rn and $^7$Be emission data used in this paper is described in Section 3.1. $^7$Be activity concentration data are available in the Radioactivity Environmental Monitoring (REM) database (https://data.jrc.ec.europa.eu/collection/id-0117). All model output, and $^{210}$Pb daily observational data at Helsinki and Sodankylä for January-March 2003 are available online (http://doi.org/10.5281/zenodo.4117521).

**Author contributions**

JA, MAHC and EB designed the study. HL and BZ conducted the GEOS-Chem model simulations. MAHC led the calculation and analysis of HYSPLIT back-trajectories. EB developed the analysis methodology and led the analysis of observational data and model output, with contributions from all coauthors. JP contributed $^{210}$Pb and meteorological observational datasets. EB wrote the manuscript with contributions from all coauthors.

**Competing interests**

The authors declare that they have no conflict of interest.

**Acknowledgements**

The paper is part of the research conducted within the project "Climate changes and their influence on the environment: impacts, adaptation and mitigation" (No. 43007) financed by the Ministry of Education, Science and Technological Development of the Republic of Serbia (2011–2019). HL and BZ acknowledge funding support from the NASA Modeling, Analysis and Prediction Program (grant 80NSSC17K0221) and Atmospheric Composition Campaign Data Analysis and Modeling program (grant NNX14AR07G). NASA Center for



Computational Sciences (NCCS) provided supercomputing resources. The GEOS-Chem model is managed by the Atmospheric Chemistry Modeling Group at Harvard University with support from NASA ACMAP and MAP programs. The University of Wyoming, Dr. Rigel Kivi from the Finnish Meteorological Institute Arctic Space Center and Dr. Laura Thölix from the Finnish Meteorological Institute are gratefully acknowledged for providing and helping with temperature profiles, ozone soundings and potential vorticity data used to support the detection

of air masses of stratospheric origin in the troposphere. The World Data Center is gratefully acknowledged for the production, preservation and dissemination of the international sunspot number (Sunspot Index and Long-term Solar Observation, SILSO, Royal Observatory of Belgium, Brussels, http://www.sidc.be/silso/datafiles#total). The NOAA/ESRL Physical Sciences Laboratory Boulder Colorado is gratefully acknowledged for providing data and plots from the Twentieth Century Reanalysis Project version 3 through their web site (http://psl.noaa.gov). Support

for the Twentieth Century Reanalysis Project version 3 dataset is provided by the U.S. Department of Energy, Office of Science Biological and Environmental Research (BER; http://science.energy.gov/ber/), by the National Oceanic and Atmospheric Administration Climate Program Office, and by the NOAA Physical Sciences Laboratory.

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





## Tables

**Table 1.** (left) Normalized differences between the MERRA-2 and observed precipitation, calculated as differences between the MERRA-2 and the observed values, normalized over the observed value, at each sampling site. Positive values indicate that the model tends to overestimate observations while the negative ones indicate underestimation. (right) Statistical parameters (mean ± SD = standard deviation; MB = mean bias; NMSE = normalized mean square error; FA2 = factor of 2) indicating the model performance in reproducing GPCP monthly accumulated precipitation at the six sampling sites in

Northern Europe.

| Sampling site | Normalized differences | | | Statistical parameters | | | | |
| | Jan-03 | Feb-03 | Mar-03 | Mean ± SD (mm) | | MB (mm) | NMSE | FA2 |
| | | | | MERRA-2 | GPCP | | | |
| **Ivalo** | -0.07 | 0.63 | 1.25 | 2.23±0.67 | 1.48±0.43 | 0.75 | 0.34 | 1 |
| **Umea** | -0.03 | 0.07 | 0.08 | 1.53±0.86 | 1.53±0.94 | 0.02 | 0 | 1 |
| **Helsinki** | -0.09 | -0.21 | 0.17 | 1.23±0.92 | 1.32±1.03 | -0.09 | 0.01 | 1 |
| **Kista** | 0.96 | -0.03 | 1.73 | 1.12±0.92 | 0.67±0.46 | 0.46 | 0.42 | 0.67 |
| **Harku** | -0.2 | 0.18 | -0.07 | 1.23±0.92 | 1.42±1.28 | -0.18 | 0.05 | 1 |
| **Risoe** | -0.06 | 2.45 | 1.52 | 1.63±0.37 | 1.05±0.98 | 0.58 | 0.3 | 0.33 |






**Table 2. (left)** Normalized differences between the simulated and observed [7]Be monthly means, calculated as differences between the simulated and the observed values, normalized over the observed value, at each sampling site. (right) Statistical parameters (mean ± SD = standard deviation; MB = mean bias; NMSE = Normalized Mean Square Error; FA2 = Factor of 2) indicating the model performance in reproducing observed [7]Be monthly means at the six sampling sites in Northern Europe

and [7]Be/[210]Pb and [210]Pb monthly means in Helsinki.

| Sampling Site | Tracer | Normalized differences | | | Statistical parameters | | | | |
|---|---|---|---|---|---|---|---|---|---|
| | | Jan-03 | Feb-03 | Mar-03 | Mean ± SD | | MB | NMSE | FA2 |
| | | | | | Modelled | Observed | | | |
| **Ivalo** | | 0.15 | 0.08 | -0.07 | (1.82±0.49) mBq m$^{-3}$ | (1.85±0.78) mBq m$^{-3}$ | -0.03 mBq m$^{-3}$ | 0.07 | 1 |
| **Umea** | | -0.01 | -0.14 | -0.05 | (1.69±0.72) mBq m$^{-3}$ | (1.88±0.96) mBq m$^{-3}$ | 1.69 mBq m$^{-3}$ | 0.89 | 1 |
| **Helsinki** | | -0.22 | -0.31 | -0.2 | (1.58±0.80) mBq m$^{-3}$ | (2.30±0.60) mBq m$^{-3}$ | 1.57 mBq m$^{-3}$ | 0.76 | 0.74 |
| **Kista** | [7]Be | -0.28 | -0.35 | -0.25 | (1.68±0.69) mBq m$^{-3}$ | (2.41±0.89) mBq m$^{-3}$ | -0.73 mBq m$^{-3}$ | 0.16 | 0.92 |
| **Harku** | | -0.21 | -0.13 | -0.28 | (1.61±0.61) mBq m$^{-3}$ | (2.16±0.81) mBq m$^{-3}$ | -0.54 mBq m$^{-3}$ | 0.17 | 0.93 |
| **Risoe** | | -0.32 | -0.4 | -0.08 | (2.08±0.83) mBq m$^{-3}$ | (3.31±1.52) mBq m$^{-3}$ | 2.08 mBq m$^{-3}$ | 0.7 | 0.62 |
| **Helsinki** | [7]Be/[210]Pb | -0.28 | -0.5 | -0.5 | 4.89±3.57 | 10.4±7.5 | -5.04 | 0.78 | 0.45 |
| **Helsinki** | [210]Pb | 0.73 | 0.5 | 0.66 | (0.48±0.29) mBq m$^{-3}$ | (0.36±0.32) mBq m$^{-3}$ | 0.12 mBq m$^{-3}$ | 0.3 | 0.78 |






**Table 3.** Statistical parameters indicating the model performance in reproducing observed [7]Be weekly (daily in the case of Helsinki) means at the six sampling sites in Northern Europe.

| Sampling site | Tracer | Mean ± SD | | MB | NMSE | R | FA2 |
|---|---|---|---|---|---|---|---|
| | | Modelled | Observed | | | | |
| Ivalo | | (1.52±0.44) mBq m⁻³ | (1.85±0.78) mBq m⁻³ | -0.33 mBq m⁻³ | 0.12 | 0.74 | 1.00 |
| Umea | | (1.43±0.72) mBq m⁻³ | (1.88±0.96) mBq m⁻³ | -0.45 mBq m⁻³ | 0.11 | 0.91 | 0.92 |
| Helsinki | [7]Be | (1.35±0.83) mBq m⁻³ | (2.30±1.15) mBq m⁻³ | -0.88 mBq m⁻³ | 0.47 | 0.64 | 0.57 |
| Kista | | (1.43±0.62) mBq m⁻³ | (2.41±0.89) mBq m⁻³ | -0.98 mBq m⁻³ | 0.30 | 0.86 | 0.62 |
| Harku | | (1.36±0.56) mBq m⁻³ | (2.16±0.81) mBq m⁻³ | -0.79 mBq m⁻³ | -0.45 | 0.73 | 0.86 |
| Risoe | | (1.84±0.89) mBq m⁻³ | (3.31±1.52) mBq m⁻³ | -1.47 mBq m⁻³ | 0.77 | 0.10 | 0.38 |
| Helsinki | [7]Be/[210]Pb | 4.86±3.96 | 10.3±7.5 | -5.06 | 0.85 | 0.66 | 0.43 |







## Figures

**Figure 1.** a) Location of the $^7$Be sampling sites in Northern Europe (source: https://mapamundiparaimprimir.com/europa/); b)
$^7$Be concentrations measured at six surface sampling sites in Northern Europe during the 2002/2003 boreal winter. Dashed
lines indicate the 90$^{th}$ percentile reference line for each sampling site.



## Total Precipitation, 2003

**Figure 2.** Comparison of the MERRA-2 total precipitation during January-March 2003 with the GPCP observations. The black dots indicate the locations of the sampling sites: 1=Ivalo, 2=Umea, 3= Helsinki, 4=Kista, 5=Harku, 6=Risoe.




**Figure 3.** Simulated monthly mean $^7$Be surface concentrations (mBq m$^{-3}$), $^7$Be/$^{210}$Pb ratio and fraction of stratospheric $^7$Be. Arrows represent winds in the MERRA-2 reanalysis. The dots indicate the locations of the sampling sites: 1=Ivalo, 2=Umea, 3=Helsinki, 4=Kista, 5=Harku, 6=Risoe.





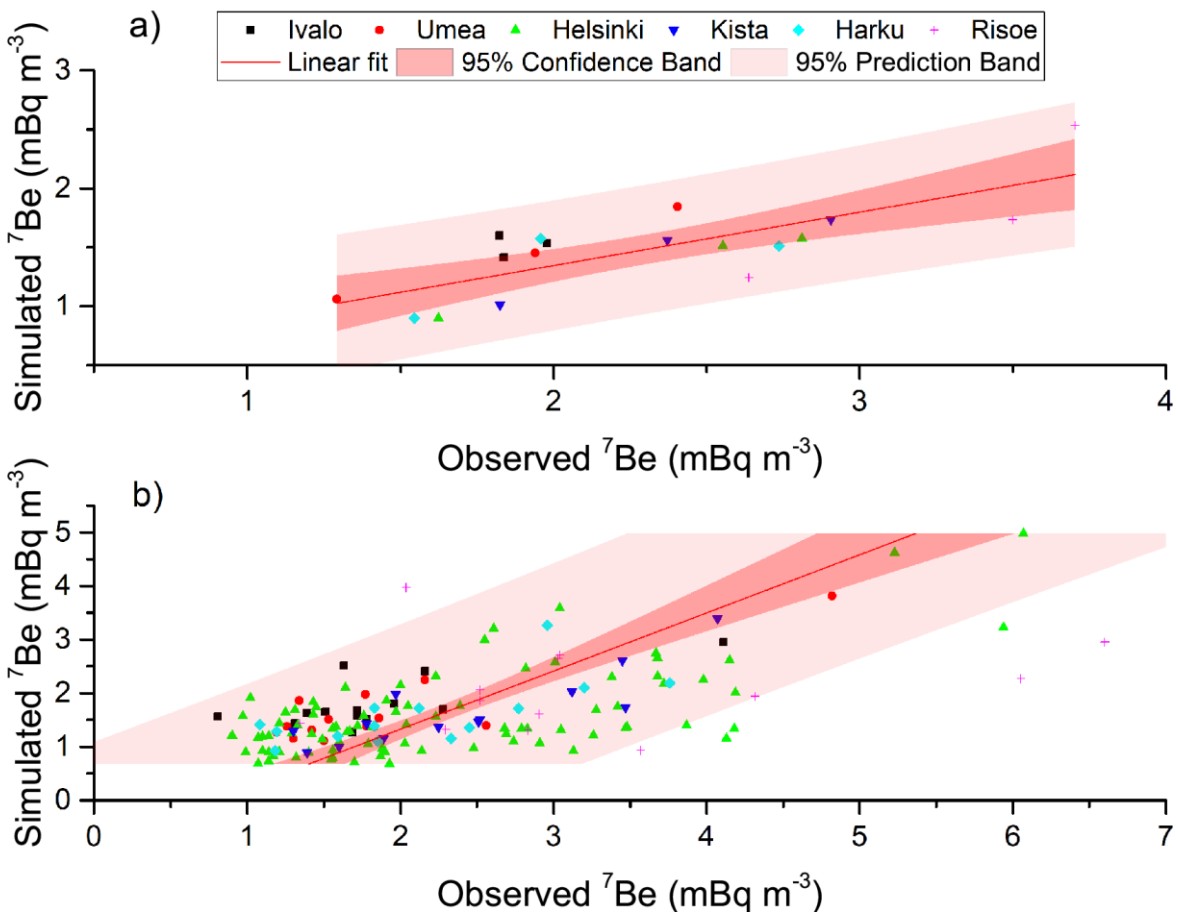

**Figure 4.** Scatter plots of: a) simulated vs. observed $^7$Be monthly means at the six sampling sites; b) simulated vs. observed $^7$Be weekly (daily in the case of Helsinki) means at the six sampling sites. Also shown are the linear fit and the 95% confidence and prediction bands around the linear fit.






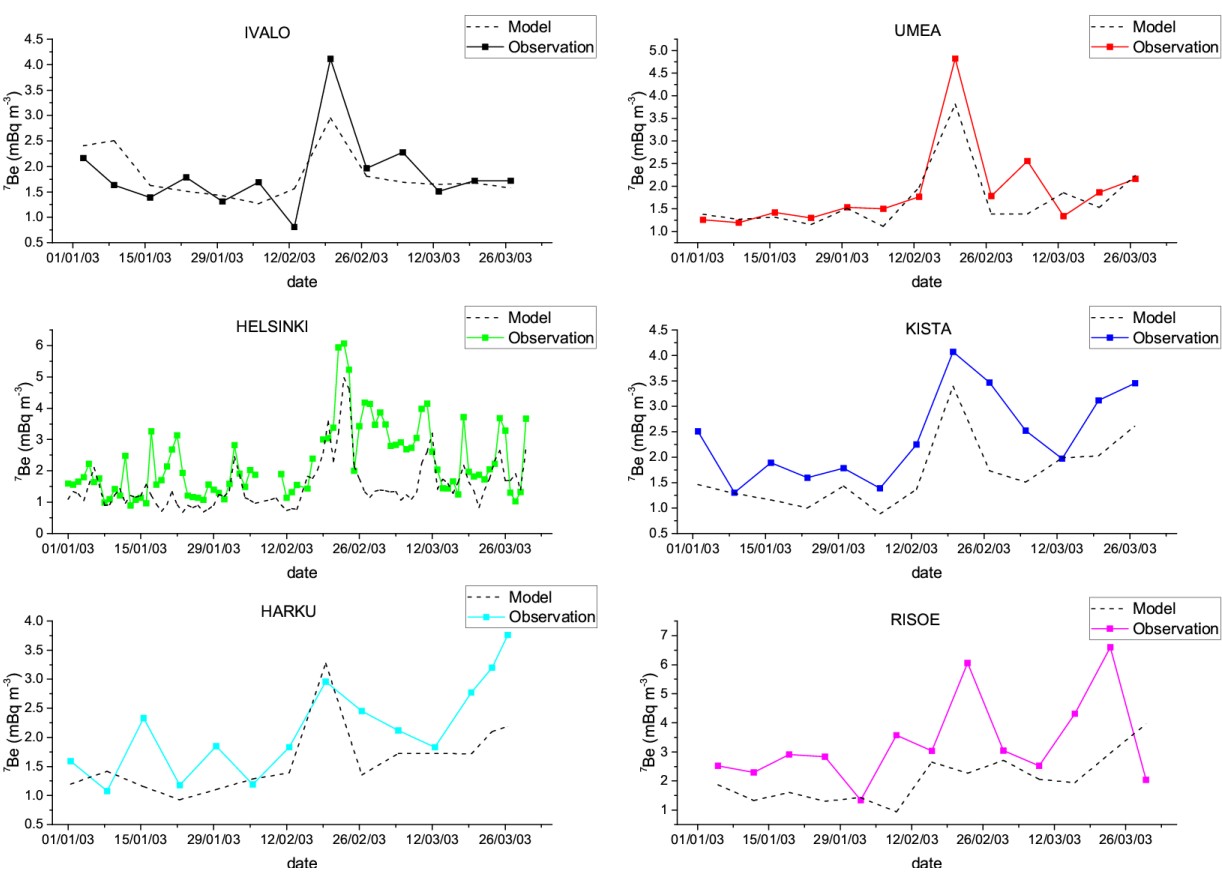

**Figure 5.** Temporal evolution of simulated and observed [7]Be surface concentrations at the six sampling sites. Values are
weekly (daily in the case of Helsinki) means.





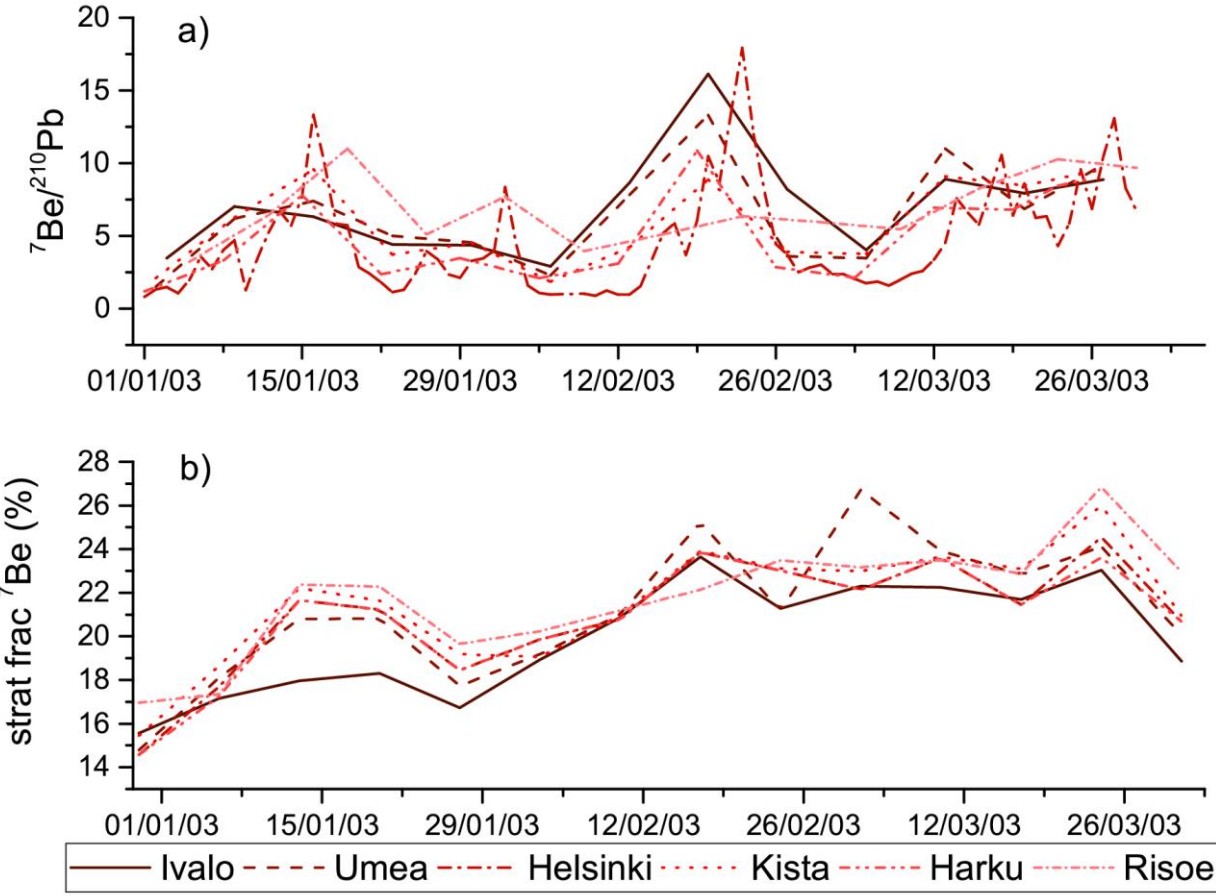

**Figure 6.** Temporal evolution of the simulated a) $^{7}Be/^{210}Pb$ and b) fraction of stratospheric $^{7}Be$ (calculated as the ratio between the stratospheric $^{7}Be$ and the total $^{7}Be$ concentrations, in percentage) at the six sampling sites during January-March 2003. Values are weekly means.







**Figure 7.** Vertical profiles of air temperature in the MERRA-2 reanalysis (dotted line) and in the soundings at the Sodankylä station in Finland on selected days of low ⁷Be values (top panels: 10 and 16 Feb 2003, 00 UTC), transition to high ⁷Be values (middle panels: 20 Feb 2003, 00 UTC and 21 Feb 2003, 12 UTC) and high ⁷Be values (bottom panels: 22 and 24 Feb 2003, 00 UTC) at the six sampling sites in Northern Europe.





**Figure 8.** Ozone soundings at the Sodankylä Arctic station during 4 different days in February 2003: 12, 19, 16 and 28 February 2003.






**Figure 9.** MERRA-2 daily mean relative humidity (colors) and winds (arrows) at ground level during February 18-25, 2003. The dots indicate the locations of the sampling sites: 1=Ivalo, 2=Umea, 3=Helsinki, 4=Kista, 5=Harku, 6=Risoe.





**Figure 10.** Time-height cross-sections of calculated daily potential vorticity during the month of February 2003 at three latitudes: a) 63°N, b) 64.5°N, and c) 66°N along the 21°E meridian.







**Figure 11.** Time-height cross-sections of simulated hourly $^{7}$Be concentrations (mBq SCM$^{-1}$ where SCM stands for Standard Cubic Meter) during the month of February 2003 at the six sampling sites.





**Figure 12.** Time-height cross-sections of MERRA-2 3-hourly average pressure vertical velocity (omega, in Pa s$^{-1}$) during the month of February 2003 sampled at the six sampling sites.



**Figure 13.** MERRA-2 daily mean pressure vertical velocity (omega) at 940 hPa during February 18-25, 2003. The dots indicate the locations of the sampling sites: 1=Ivalo, 2=Umea, 3=Helsinki, 4=Kista, 5=Harku, 6=Risoe.





**Figure 14.** Simulated daily mean fraction of $^7$Be originating in the stratosphere (%) at 940 hPa. The dots indicate the locations
815   of the sampling sites: 1=Ivalo, 2=Umea, 3=Helsinki, 4=Kista, 5=Harku, 6=Risoe.





**Figure 15.** Average trajectory cluster results (centroids) in Ivalo, Harku and Risoe at 1000 m for low $^7$Be values (left: 3-16 February 2003) and high $^7$Be values (right: 20-28 February 2003), respectively. The stations are ordered by latitude from top to bottom (coordinates of the receptor site are provided on the left of each plot). The right numbers in the centroids are the percentage of complete trajectories occurring in that cluster, and the left numbers are an identification number of the centroid.