# Peer review of "Observation and modeling of surface high-7Be concentration events in Northern Europe associated with the instability of the Arctic polar vortex in early 2003"

_Atmospheric Chemistry and Physics, 2020_

## Referee Comment (RC2)

**Review of "Observation and modelling of high-7Be "**

BY E. BRATTICH ET AL.

**General remarks**

This paper presents a case study of high $^7$Be events observed in Northern Europe in early 2003. The authors use a model simulation to interpret the presented measurements. They further consider a range of auxiliary parameters like potential vorticity. These data are of interest to the readership of ACP and the presented analysis helps the understanding of the measurements.

However, some aspects are not discussed. I think the altitude of the measurement sites should be mentioned (I have not looked them up). Transport from the stratosphere is considered to be an important question, but how far down reaches the transport? This is not an obvious issue as not all stratospheric intrusions reach sea-level. Also, are there any ground level ozone measurements at these sites that could be considered? Ozone profiles are considered (Fig. 8), but here I cannot see a huge enhancement. Of course ozone has different properties than $^7$Be, but such issues could be better discussed.

Further, the agreement of the model results and the observations need not to be perfect; this is a different modelling task. But there should be more focus on the *processes* that cause the downward transport. Such an analysis should go beyond stating that there is reasonable agreement between the model and the observations. Figure 7 shows temperature profiles that show differences regarding the days when high $^7$Be events are observed, but what do we learn about the processes at work? Is there anything special about the SSW in early 2003 (on which the paper focuses)? Or would one expect such events for all SSWs that regularly occur in Arctic winter? Is there any altitude dependence of the stratospheric intrusion; how well is (or needs to be) the planetary boundary layer simulated (for the simulation of an intrusion down to the ground). All these questions are not addressed in the manuscript. While I do not expect that such questions can all be satisfactorily answered, I suggest more discussion of the issues. And more focus on the processes of

stratospheric intrusions under conditions of a SSW.

Regarding the organisation of the paper, I note that it has many sections, but no section called "Results"; perhaps such a section could be introduced, with the appropriate subheadings. There are also some issues with the wording and with the references (I have listed some examples below), so I suggest a careful proofreading of the revised version. (By the way, the references in this review are only to make the points I mention more easily understandable; this is not a suggestion for citations).

In summary, I think this is an interesting paper, but it needs a revision addressing better the points raised above). I would expect a revised version of this paper would be acceptable and would be of interest to the readers of ACP.

**Comments in detail**

- Title: this paper is about surface observations; you could introduce this word in your title

- l. 15: Say at which altitudes these values are recorded.

- l. 30: radiative or radioactive?

- l. 37: I think instead of vertical transport you mean downward transport

- l. 49/50: The study of Salminen-Paatero is certainly not the using PV to analyse transport of air into the troposphere. I suggest to be more specific on Salminen-Paatero or to discuss the aspect more generally, which would likely involve more references. A classic paper is for example Danielsen (1968) an there is the review by Holton et al. (1995, cited in the paper elsewhere).

- l. 61: change to: and the Southern hemisphere

- l. 63: drop 'the' and 'seasons'

- l. 65: There is a lot of discussion in the paper on SSWs; given the importance of the concept and in particularly the extension of the impact to outside the polar vortex I suggest a bit more discussion on SSWs

(e.g., Charlton and Polvani, 2007; Charlton et al., 2007; Sofieva et al., 2012; Tao et al., 2015)

- l. 87: Many people (including me) would argue that only the Antarctic ozone hole should be called by this name, even very strong recent Arctic ozone losses (Manney et al., 2020) have not been refered to as an "ozone hole". Suggest changing the wording.

- l. 129: The 7Be data are discussed here. I suggest in addition a short explanation on the measurement principle.

- l. 159: which data set from ECMWF was used? Perhaps ERA-I (Dee et al., 2011)? Be specific and provide a citation. Also add ECMWF to the acknowledgements.

- l. 195: which vertical velocity was used in the HYSPLIT calculations?

- l. 205: For which altitudes is this statement relevant and appropriate? E.g. for the entire troposphere?

- l. 243: suggest 'chemical composition' (if this is what is meant here.

- l. 255: An alternative data set is TRMM: why is this data set not considered? Not the right region? You could briefly comment. Also: there might be local precipitation measurements at the sites in question.

- l. 289: is the horizontal or the vertical resolution the issue here?

- l. 334: not really clear, I think you mean something like "impact on surface weather"

- l. 355: Say "MERRA-2" her, this is not the same thing as a simulation

- l. I agree that the temperature structure is different, but where in these profiles do I see an indication for the processes causing downward transport?

- l. 370: ozone in the troposphere – is this shown here? Do you need a reference?

- l. 374: there is nothing wrong with using potential vorticity from ECMWF, however potential vorticity can also be computed from MERRA-2 data, which might be more consistent. Is there a reason for using ECMWF here?

- l. 377: potential vorticity is not really a conserved quantity; what is the life time in the troposphere? In this way one could learn about the timescales of vertical transport

- l. 389: why 1000 m? This model assumption should be explained at this point. y

- l. 398: Give the exact reference (fig., section) where this is shown.

- l. 414: Perhaps say more clearly that the *horizontal* resolution is your point here.

- l. 422/423: I agree that tropospheric ozone on 19 February is somewhat enhanced, but not very much (30 ppb is not a high value. And the ozone signal does not reach the ground; is this in line with you explanations?

- l. 434: here and above; you use the backward trajectories for a discussion of the vertical transport, which is the key issue here. Why is this information not considered?

- l. 440: in a future climate there might be more SSWs – but would all of these SSWs lead to high surface 7Be values? I think you need to argue for more SSWs of the type considered here.

- l. 442/443: You should state here, where the data are *available*, not where in the paper they are discussed.

- l. 690: I think the journal should be abbreviated here

- l. 692: 2017 or 2016? Both years are in the reference. . .

- Fig. 1: here and elsewhere, I think the town in Sweden is called Umeå, use \aa in LATEXif you like.

- Fig. 1: remove grey background from bottom panel.

- Fig 12: the black line is the zero line – correct? Mention in the caption.

**References**

Charlton, A. J. and Polvani, L. M.: A new look at stratospheric sudden warmings. Part I: Climatology and modeling benchmarks, Journal of Climate, 20, 449–469, 2007.

Charlton, A. J., Polvani, L. M., Perlwitz, J., Sassi, F., Manzini, E., Shibata, K., Pawson, S., Nielsen, J. E., and Rind, D.: A new look at stratospheric sudden warmings. Part II: Evaluation of numerical model simulations, Journal of Climate, 20, 470–488, 2007.

Danielsen, E. F.: Stratospheric-tropospheric exchange based on radioactivity, ozone and potential vorticity, J. Atmos. Sci., 25, 502–518, 1968.

Dee, D. P., Uppala, S. M., Simmons, A. J., Berrisford, P., Poli, P., Kobayashi, S., Andrae, U., Balmaseda, M. A., Balsamo, G., Bauer, P., Bechtold, P., Beljaars, A. C. M., van de Berg, L., Bidlot, J., Bormann, N., Delsol, C., Dragani, R., Fuentes, M., Geer, A. J., Haimberger, L., Healy, S. B., Hersbach, H., Hólm, E. V., Isaksen, L., Kållberg, P., Köhler, M., Matricardi, M., McNally, A. P., Monge-Sanz, B. M., Morcrette, J.-J., Park, B.-K., Peubey, C., de Rosnay, P., Tavolato, C., Thépaut, J.-N., and Vitart, F.: The ERA-Interim reanalysis: configuration and performance of the data assimilation system, Q. J. R. Meteorol. Soc., 137, 553–597, https://doi.org/10.1002/qj.828, 2011.

Holton, J. R., Haynes, P., McIntyre, M. E., Douglass, A. R., Rood, R. B., and Pfister, L.: Stratosphere-troposphere exchange, Rev. Geophys., 33, 403–439, 1995.

Manney, G. L., Livesey, N. J., Santee, M. L., Froidevaux, L., Lambert, A., Lawrence, Z. D., Milln, L. F., Neu, J. L., Read, W. G., Schwartz, M. J., and Fuller, R. A.: Record-Low Arctic Stratospheric Ozone in 2020: MLS Observations of Chemical Processes and Comparisons With Previous Extreme Winters, Geophys. Res. Lett., 47, e2020GL089063, https://doi.org/https://doi.org/10.1029/2020GL089063, 2020.

Sofieva, V., Kalakoski, N., Verronen, P., Päivärinta, S.-M., Kyrölä, E., Backman, L., and Tamminen, J.: Polar-night $O_3$, $NO_2$ and $NO_3$ distributions during sudden stratospheric warmings in 2003–2008 as seen by GOMOS/Envisat, Atmos. Chem. Phys., 12, 1051–1066, 2012.

Tao, M., Konopka, P., Ploeger, F., Grooß, J.-U., Müller, R., Volk, C., Walker, K., and Riese, M.: Impact of the 2009 major stratospheric sudden warming on the composition of the stratosphere, Atmos. Chem. Phys., pp. 8695–8715, https://doi.org/10.5194/acp-15-8695-2015, 2015.

---

## Author Response (AR1)

**Reply to the reviewers' comments on "Observation and modeling of high-[7]Be events in Northern Europe associated with the instability of the Arctic polar vortex in early 2003" by Erika Brattich et al.**

Manuscript Ref: acp-2020-1121

We thank the reviewers for their useful comments. Below are the reviewers' comments in italic followed by our replies in blue text.

*RC1*

*The manuscript presents the results of an analysis of the atmospheric conditions, in particular, a stratospheric sudden warming (SSW) event in late February 2003, leading to an observed increase of 7Be concentration in the near-ground air. The qualitative analysis is comprehensive, and the association between the SSW event and the observed 7Be increase is convincing and worth publishing. However, the quantitative model contains a serious flaw and needs to be corrected before the manuscript becomes acceptable since the modelled 7Be concentrations cannot be trusted.*

*This reviewer was deeply surprised by the rough and inappropriate way the production of 7Be was modelled. The authors state that they used production estimated by Lal & Peters (1967, called LP67 here) for 1958 (was it based in Fig. 20 there?), which is unacceptable for several reasons: i) the model of LP67 is greatly outdated as based on a very rough and approximate approach (an analytically estimated rate of nuclear "stars" in the atmosphere converted with the mean production yield of 7Be per star). This approach is quite uncertain as compared to modern full Monte-Carlo simulations of the cosmic-ray-induced atmospheric nucleonic cascade. Instead, the most recent and accurate production model by Poluianov et al. (2016, doi: 10.1002/2016JD025034), based on the GEANT-4 Monte-Carlo tool, is highly recommended for use. Comparing to the full Monte-Carlo model, the results by LP67 OVERESTIMATE the 7Be production by 30-50% (cf. Tab.3 of LP67 and Tab.1 of Poluianov et al., 2016). ii) The level of solar activity and the corresponding modulation of cosmic rays (hence 7Be production) in 1958 was significantly higher than that in 2003, as the authors realize (see line 295). Accordingly, by applying the 1958 production to 2003, the authors UNDERESTIMATE the production. iii) The authors ignore the change of the geomagnetic field strength, which was reduced by ~4% between 1958 and 2003. In this way, they also slightly UNDERESTIMATED 7Be production. Altogether, the three errors work in opposite directions making the quantitative result unreliable. The authors are requested to redo modelling using an appropriate 7Be production model. In case this would require too much work, the authors can make a compromise: the present LP67-based model results can be scaled to the correct global (or polar) production estimated by an appropriate model. However, this would be only an approximate temporal solution. In all further works, the authors are required to use a relevant production model. Before this flaw is corrected, the manuscript cannot be accepted for publication.*

We thank the reviewer for his/her comments and suggestions. We have improved the discussion on the use of the Lal and Peters (1967) [7]Be production rates in the revised version at lines 193-209. Indeed, published estimates of [7]Be production rates (Lal and Peters, 1967; O'Brien et al., 1991; Masarik and Reedy, 1995; Masarik and Beer, 1999; Usoskin and Kovaltsov, 2008; Poluianov et al., 2016) greatly differ, with global mean column production rates ranging over an average solar cycle from 0.035 atoms cm$^{-2}$ s$^{-1}$ (Masarik and Beer,

1999), 0.063 atoms cm$^{-2}$ s$^{-1}$ (O'Brien et al., 1991), to 0.081 atoms cm$^{-2}$ s$^{-1}$ (Lal and Peters, 1967). Previous studies (e.g., Koch et al., 1996; Liu et al., 2001) found that using the O'Brien et al. (1991) $^7$Be source (0.063 atoms cm$^{-2}$ s$^{-1}$) in global models yields a consistent underestimate of observed surface and stratospheric $^7$Be concentrations. The $^7$Be production rates provided by Usoskin and Kovaltsov (2008) and Poluianov et al. (2016), with global mean values of 0.062 atoms cm$^{-2}$ s$^{-1}$ and 0.065 atoms cm$^{-2}$ s$^{-1}$, respectively, are broadly consistent with those of O'Brien et al. (1991)) and are about 25% lower than those of Lal and Peters (1967). We use the $^7$Be production rates recommended by Lal and Peters (1967) for a maximum solar activity year (1958), which has been shown to produce the best results compared to aircraft $^7$Be observations in the stratosphere where $^7$Be concentrations mainly result from a balance between production and radioactive decay and their observations can be used as a constraint on the $^7$Be source (Koch et al., 1996; Liu et al., 2001). Although using the most recent production rates of Poluianov et al. (2016) is planned for a future modeling study (e.g., Golubenko et al., 2021), using those rates would likely result in model $^7$Be concentrations biased low, especially in the stratosphere as aforementioned. In addition, it is noted that the focus of this paper is not on the perfect agreement between model simulations and observations, but rather on the use of model simulations to investigate the processes responsible for the $^7$Be peak observed at different northern latitude stations in Fennoscandia in early 2003.

*Other minor comments and suggestions are listed below:*
*1)   The title would sound more correctly if "high-7Be events" was replaced with "high 7Be concentration events".*
We thank the reviewer for the suggestion. The title was changed accordingly.

*2)   It would be worth to refer to previous works on full atmospheric dynamical models applied for studies of 7Be transport/deposition: the ECHAM-HAM5 (Heikkilä et al., ACP, 2008, doi: 10.5194/acp-8-2797-2008) and the GISS model (Usoskin et al., JGR, 2009, doi: 10.1029/2008JD011333)*
We thank the reviewer for this suggestion. Both references are now cited in Introduction (lines 35-40).

*3)                Line       32:       "stratospheric       influence"       on       what?*
We have revised the sentence to "$^7$Be is considered a tracer for intrusion of stratospheric air to the troposphere and large-scale subsidence (e.g., Liu et al., 2016; Chae and Kim, 2019; Heikkila et al., 2008)".

*4)   Line 193: "previously archived restart files" – please specify what it is.*
We have revised the text at lines 229-231 to "All model simulations are conducted for the period of January 2002 – March 2003 with initial conditions from a previous five-year simulation. Hourly and monthly mean outputs for January-March 2003 are used for analysis."

*5)   Line 218: NMSE does not provide an estimate of whether the difference is statistically significant or not. Z-test is recommended instead, which gives a measure of the statistical significance of the difference.*
Agreed. The evaluation of the Z-test has been added to Table 3 and discussed in lines 366-368.

*6)   Line 223: the statistical significance of the correlation coefficient should be evaluated. With so short*

*analyzed series, even a high correlation coefficient can be insignificant.*
Agreed. All values but the one at Risø were significant, and this information has been added to Table 3.

*7)    Line 273: the correlation of -0.32 for Ivalo implies a failure. This needs an explanation.*
An explanation for this observation was added in lines 316-321: "The low negative correlation at Ivalo is due to the fact that while the GPCP-observed precipitation at this site is similar between January and February with a general tendency towards lower values from January to March 2003, the model simulates a decrease from January to February with a small increase in March. However, the statistical parameters reported in Table 1 indicate that an overall small discrepancy between the GPCP and MERRA-2 precipitation at all sites."

*8)   Line 296: reference to O'Brien (1979) is not appropriate here. A review by Potgieter (Liv. Rev. Sol. Phys., 2013,         doi:         10.12942/lrsp-2013-3)         is         recommended         instead.*
The reference was modified accordingly.

*9)    Finland is not a part of Scandinavia. The analyzed region should be called Fennoscandia.*
We thank the reviewer for this suggestion. The revised version now properly refers to Fennoscandia instead of Scandinavia.

*10)  The term of the stratospheric fraction of 7Be needs to be strictly defined. Presently, it is presented as the ratio of stratospheric to global concentrations, which is vague. Are these concentrations mean global or polar regions, for what period (tropopause height varies in time). Please provide a formula.*
We have revised the text (lines 225-229) to "In addition to the standard model simulations of $^7$Be and $^{210}$Pb, we separately transport $^7$Be produced in the model layers above the MERRA-2 thermal tropopause (i.e., stratospheric $^7$Be tracer) to quantify the stratospheric contribution to $^7$Be in the troposphere. This approach was previously used by Liu et al. (2001, 2016). Stratospheric fraction of $^7$Be is defined as the ratio of the stratospheric $^7$Be tracer concentration to the $^7$Be concentration from the standard simulation."

*11)  Line 413: the model tends to underestimate 7Be concentrations – see the major concern above.*
Please see our response to the major comment above.

*12)  Figure 1b: for what periods were the 90% levels defined? The upper dotted line rises many questions: only two points lie above it, how many are overall? Why does the line lie between points? Dotted lines need to be marked as to which station they correspond to. It is recommended that the points are connected, otherwise, it    is    hardly    possible    to    distinguish    data    from    different    sites.*
The 90$^{th}$ percentile levels were defined over the 1995-2011 period, and peak events were defined as values exceeding this threshold. This information was added to the revised version of the manuscript (lines 276-277). In this sense, the time series in Fig. 1b shows only a three-month interval (January, February and March 2003) of the whole 1995-2011 period, with a small number of peak $^7$Be events, which are the subject of this study. For instance, at Risø, the procedure allowed to detect a total number of 25 peak events, of which only one was recorded in the boreal winter 2003 period. The figure was revised to contain the information of the sampling site corresponding to each 90$^{th}$ percentile line, and the points are now connected.

*13)  Fig.4b looks strange. This reviewer would place a linear fit at a shallower slope. Can the authors specify*

*how             the             fit            was          obtained?*

Now we state in the Figure 4 caption that "Also shown are the linear regression line and …". It looks strange probably because the two plots are above each other and might "trick the eye". Indeed, when the plot is reproduced alone, the fit looks "normal" (see below).

[Figure]

*14) Fig. 6. The authors are advised to use different colours for the lines. Also, the absence of a peak in Risoe data is worth more discussions.*

The colours were modified to meet the request from the reviewer as well as the Editorial team to consider colour blindness in the colour choices. Additional discussions for the absence of the $^{7}Be/^{210}Pb$ peak in Risø data were added at lines 400-402.

*RC2*

*General remarks*

*This paper presents a case study of high 7Be events observed in Northern Europe in early 2003. The authors use a model simulation to interpret the presented measurements. They further consider a range of auxiliary*

*parameters like potential vorticity. These data are of interest to the readership of ACP and the presented analysis helps the understanding of the measurements.*

We thank the reviewer for his/her comments.

*However, some aspects are not discussed. I think the altitude of the measurement sites should be mentioned (I have not looked them up). Transport from the stratosphere is considered to be an important question, but how far down reaches the transport? This is not an obvious issue as not all stratospheric intrusions reach sea-level. Also, are there any ground level ozone measurements at these sites that could be considered? Ozone profiles are considered (Fig. 8), but here I cannot see a huge enhancement. Of course ozone has different properties than 7Be, but such issues could be better discussed.*

We thank the reviewer for these comments. The revised version of the manuscript now contains information on the elevations of measurement sites (lines 153-154). All the measurement sites are located at low elevations (maximum 130 m a.s.l. for the site of Ivalo). We have added discussions of surface ozone in the text (lines 412-416): "In addition, average $O_3$ values recorded at surface air quality stations located in Denmark, Finland, and Sweden, which are available through the saqgetr R package (Grange, 2019), show enhanced $O_3$ concentrations in late February 2003, consistent with the aforementioned peaks in the $^7$Be/$^{210}$Pb ratio as well as stratospheric $^7$Be fraction. This further suggests the transport of stratospheric air masses to the surface."

*Further, the agreement of the model results and the observations need not to be perfect; this is a different modelling task. But there should be more focus on the processes that cause the downward transport. Such an analysis should go beyond stating that there is reasonable agreement between the model and the observations. Figure 7 shows temperature profiles that show differences regarding the days when high 7Be events are observed, but what do we learn about the processes at work? Is there anything special about the SSW in early 2003 (on which the paper focuses)? Or would one expect such events for all SSWs that regularly occur in Arctic winter? Is there any altitude dependence of the stratospheric intrusion; how well is (or needs to be) the planetary boundary layer simulated (for the simulation of an intrusion down to the ground). All these questions are not addressed in the manuscript.*

We agree with the reviewer that the paper is not focused on model simulations but rather on the processes that cause the downward transport. The revised version of the manuscript (Introduction, lines 88-99) contains a deeper discussion on the processes occurring in early 2003 and on the evidences that link this SSW with an intense transport of stratospheric air masses to the ground. "Previous works suggested that the occurrence of SSWs perturb greatly the polar vortex and hence the stratospheric PV distribution (e.g., Matthewman et al., 2009) and the vertical distribution of ozone (e.g., Sonneman et al., 2006; Madhu, 2016). Additionally, the meteorological conditions associated with SSWs in the Arctic have been linked with the occurrences of $^7$Be winter extremes, especially in the presence of a very high Scandinavian teleconnection index (Ajtic et al., 2018)."

*While I do not expect that such questions can all be satisfactorily answered, I suggest more discussion of the issues. And more focus on the processes of stratospheric intrusions under conditions of a SSW.*

*Regarding the organisation of the paper, I note that it has many sections, but no section called "Results"; perhaps such a section could be introduced, with the appropriate subheadings. There are also some issues with the wording and with the references (I have listed some examples below), so I suggest a careful*

*proofreading of the revised version. (By the way, the references in this review are only to make the points I mention more easily understandable; this is not a suggestion for citations).*

We have added a section called "Results and discussion" (Section 4), which includes subsections 4.1-4.5 presenting the analysis of model simulations and the interpretations of observed [7]Be variabilities.

*In summary, I think this is an interesting paper, but it needs a revision addressing better the points raised above). I would expect a revised version of this paper would be acceptable and would be of interest to the readers of ACP.*

We thank the reviewer for his/her comments. In the following we provide a point-by-point response.

***Comments in detail***
*_ Title: this paper is about surface observations; you could introduce this word in your title*

Agreed. The word "surface" has been added in the title.

*_ l. 15: Say at which altitudes these values are recorded.*

Agreed. We have revised the sentence (lines 16-17) to "Events of very high concentrations of [7]Be cosmogenic radionuclide have been recorded at low-elevation surface stations in the subpolar regions of Europe during the cold season.". In addition, information on the elevation range of the sampling sites was added at lines 153-154.

*_ l. 30: radiative or radioactive?*

Changed to "radioactive".

*_ l. 37: I think instead of vertical transport you mean downward transport*

Yes, the phrase has been changed to "downward transport in the troposphere".

*_ l. 49/50: The study of Salminen-Paatero is certainly not the using PV to analyse transport of air into the troposphere. I suggest to be more specific on Salminen-Paatero or to discuss the aspect more generally, which would likely involve more references. A classic paper is for example Danielsen (1968) and there is the review by Holton et al. (1995, cited in the paper elsewhere).*

Thanks for the suggestion. It is true that the study of Salminen-Paatero et al. (2019) is not focused on potential vorticity but rather on radionuclides, and uses potential vorticity to gain insights into stratosphere-to-troposphere transport. This information was added in the revised version of the manuscript. We have revised the sentence to "A recent study by Salminen-Paatero et al. (2019) who used potential vorticity analysis to gain insights into stratosphere-to-troposphere transport of radionuclides at Rovaniemi (Finnish Lapland) indicated that the transfer of stratospheric air into the troposphere was at its maximum in March followed by gradual movement into the ground-level during spring and early summer."

*_ l. 61: change to: and the Southern hemisphere*

Changed accordingly.

*_ l. 63: drop `the' and `seasons'*

Changed accordingly.

_l. 65: There is a lot of discussion in the paper on SSWs; given the importance of the concept and in particularly the extension of the impact to outside the polar vortex I suggest a bit more discussion on SSWs (e.g., Charlton and Polvani, 2007; Charlton et al., 2007; Sofieva et al., 2012; Tao et al., 2015)_

Thanks for the suggestion. The following discussion and references on SSWs were added/updated in the revised version of the manuscript. "A higher temporal variability of the Arctic vortex includes the SSW, the strongest manifestation of the coupling of the stratosphere-troposphere system, with influence on the tropospheric flow lasting for many weeks (Charlton and Polvani, 2007) and with significant effects on chemical composition in the middle atmosphere (Sofieva et al., 2011; Tao et al., 2015). While major SSWs, the so-called vortex split (Charlton and Polvani, 2007; Charlton et al., 2007) can even cause the stratospheric vortex to break down during midwinter (Waugh et al., 2017), vortex displacements are instead characterized by a shift of the polar vortex off the pole and its subsequent distortion into a "comma shape" during the extrusion of a vortex filament (Charlton and Polvani, 2007; Charlton et al., 2007)."

_l. 87: Many people (including me) would argue that only the Antarctic ozone hole should be called by this name, even very strong recent Arctic ozone losses (Manney et al., 2020) have not been refered to as an "ozone hole". Suggest changing the wording._

Thanks for pointing this out. We have revised the sentence to "While the initial scientific interest over the stratospheric polar vortex was especially linked to the stratospheric ozone loss over the poles, it is now recognized that the vortices might affect the processes in the troposphere and surface weather (e.g., Mitchell et al., 2013)."

_l. 129: The 7Be data are discussed here. I suggest in addition a short explanation on the measurement principle._

The following explanation of the measurement principle was added: "In particular, $^7$Be activity concentrations were obtained by gamma-spectrometry analysis performed by the European Union Competent Authorities. Aerosol samples were collected on filter papers using air samplers with a flow rate of several hundred cubic meters per day and then their radioactivity concentrations were analysed in laboratories."

_l. 159: which data set from ECMWF was used? Perhaps ERA-I (Dee et al., 2011)? Be specific and provide a citation. Also add ECMWF to the acknowledgements._

We have added the following information and reference about the ERA-Interim data set used to calculate potential vorticity. "To study the effect of downward transport …, potential vorticity (PV) values (Holton et al., 1995) were calculated from ERA-Interim wind, temperature, and surface pressure fields (Dee et al., 2011) …". ECMWF was also added in the Acknowledgements.

_l. 195: which vertical velocity was used in the HYSPLIT calculations?_

We have added the following sentence on the vertical velocity used in the HYSPLIT calculations: "Computation used the vertical velocity field (https://www.ready.noaa.gov/documents/Tutorial/html/traj_vert.html) contained in the meteorological input file."

_l. 205: For which altitudes is this statement relevant and appropriate? E.g. for the entire troposphere?_

The statement at lines 243-245 ("It is worth mentioning that clusters, as well as trajectories, indicate an estimation of the general airflow rather than the exact pathway of an air parcel (e.g., Jorba et al., 2004; Salvador et al., 2008).") is rather general for trajectories, and not specific to a certain altitude. No change was made here.

_ l. 243: suggest `chemical composition' (if this is what is meant here.
Revised accordingly.

_ l. 255: An alternative data set is TRMM: why is this data set not considered? Not the right region? You could briefly comment. Also: there might be local precipitation measurements at the sites in question.
The Tropical Rainfall Measuring Mission (TRMM) provides data on precipitation in the tropical and subtropical regions of the Earth, i.e., not in the high-latitude region of this study. We have opted not to cite this data set to avoid confusion, while a reference to local ECA&D data and its comparison with the MERRA-2 reanalysis was already provided in Section 4.1 in the original manuscript.

_ l. 289: is the horizontal or the vertical resolution the issue here?
We have changed the phrase to "coarse horizontal resolution of the model" in the revised version of the manuscript.

_ l. 334: not really clear, I think you mean something like "impact on surface weather"
The word "surface" was added before "weather conditions" in the revised version of the manuscript.

_ l. 355: Say "MERRA-2" her, this is not the same thing as a simulation
"simulated" was changed to "MERRA-2".

_ l. I agree that the temperature structure is different, but where in these profiles do I see an indication for the processes causing downward transport?
We did not use the temperature profile as the main evidence of downward transport for the enhanced [7]Be event on Feb. 24, 2003. As stated earlier in this paragraph, SSW suggested disturbance (mixing) in the stratosphere, which might cause downward transport to the troposphere and enhanced [7]Be concentration at the surface sites. This link is then reinforced by the contemporary reversal of zonal winds. Therefore, no further change has been made as regards this comment.

_ l. 370: ozone in the troposphere - is this shown here? Do you need a reference?
Figure 8 (a) shows ozone soundings up to 15 km, i.e., in the troposphere and lower stratosphere. In this sense, the reference to Figure 8 here is correct.

_ l. 374: there is nothing wrong with using potential vorticity from ECMWF, however potential vorticity can also be computed from MERRA-2 data, which might be more consistent. Is there a reason for using ECMWF here?
We agree that the use of MERRA-2 data is more consistent. Vertical cross sections of potential vorticity at the six sampling sites, which are overall in good agreement with those using ECMWF as presented in the main text, were added in the revised Supplementary Material.

_ l. 377: potential vorticity is not really a conserved quantity; what is the life time in the troposphere? In this way one could learn about the timescales of vertical transport

It is true that potential vorticity is not really a conserved quantity. In fact, potential vorticity is conserved for adiabatic and inviscid flow (Gettelman et al., 2011). Indeed, from a dynamical point of view, irreversible changes across the tropopause require a change in potential vorticity, which in turn require the presence of diabatic processes (e.g., Shapiro, 1980; Hoor et al., 2010). However, because of its conservation properties under adiabatic conditions, potential vorticity is considered a quasi-passive tracer with the crucial distinction from chemical tracers that it is not just simply advected by the flow, but induces the flow at the same time (Hoskins et al., 1985; Gettelman et al., 2011). On a given isentrope, the tropopause level is identified by regions of strong enhancement in potential vorticity gradients, with well distinct values in the troposphere and in the stratosphere. The lifetime of potential vorticity cutoff lows is still a subject of a great debate, as testified by the increasingly high number of papers on this subject. Recent papers (e.g., Portmann et al., 2020; Pinheiro et al., 2017; Muñoz et al., 2020) suggest that most events in the extratropics and in the Northern Hemisphere are relatively short-lived, persisting for about 2-3 days, with rare more persistent events. Part of this discussion has been added in the revised version of the paper (lines 441-446).

_ l. 389: why 1000 m? This model assumption should be explained at this point.

We chose to only study the results above winter PBL height. We have stated in the text that results at other altitudes in the lower troposphere are similar.

_ l. 398: Give the exact reference (_g., section) where this is shown.

There is no reference to provide here, as the downward transport of stratospheric air was previously identified in Section 4.1 in the revised manuscript.

_ l. 414: Perhaps say more clearly that the horizontal resolution is your point here.

We have changed the phrase to "due to its coarse horizontal resolution".

_ l. 422/423: I agree that tropospheric ozone on 19 February is somewhat enhanced, but not very much (30 ppb is not a high value. And the ozone signal does not reach the ground; is this in line with you explanations?

We thank the reviewer for his/her comment. Indeed, the ozone sounding of 19th February shows a small enhancement in tropospheric ozone, and the signal reaches 1-km above ground level. This observation may seem somewhat not consistent with the $^7$Be peak. To better elucidate the link between SSW and downward transport of $^7$Be-O$_3$ rich air from the stratosphere, the revised manuscript also contains a plot of mean O$_3$ values (new Figure 8, panel b) (available as open data from the saqgetr package) recorded at ground-based air quality stations in Denmark, Sweden and Finland, which shows a contemporary enhancement in late February 2003, better linked with the $^7$Be peak, demonstrating more clearly the connection between the $^7$Be enhancement and downward transport from higher altitudes.

[Figure]

Figure 8. a) Vertical profiles of ozone mixing ratios (ppbv) obtained by ozone soundings at the Sodankylä Arctic station during 4 different days in February 2003: 12, 19, 16 and 28 February 2003; b) Daily mean $O_3$ concentrations recorded at ground-based air quality stations located in Denmark (DK), Finland (FI), Sweden (SE) during January-March 2003.

_ l. 434: here and above; you use the backward trajectories for a discussion of the vertical transport, which is the key issue here. Why is this information not considered?

Back-trajectories are used to provide information on the different circulation patterns during the study period, and in particular to suggest the reversal of zonal winds to airflows from upper vertical levels during the [7]Be peak period. In this sense, as an independent approach, the clusters of back-trajectories improve the understanding of the kind of transport responsible for the [7]Be peak. Further reference to the use of clusters of back-trajectories to analyze the circulation pattern is given at lines 241-243.

_ l. 440: in a future climate there might be more SSWs - but would all of these SSWs lead to high surface 7Be values? I think you need to argue for more SSWs of the type considered here.

We agree that for sure not all SSWs will lead to high [7]Be surface concentrations, but here we would like to point out the general implication of our results in the broader context of climate change.

_ l. 442/443: You should state here, where the data are available, not where in the paper they are discussed.

Agreed. The sentences on the sections where the data are discussed were removed.

_ l. 690: I think the journal should be abbreviated here

Done.

_ l. 692: 2017 or 2016? Both years are in the reference…

It is 2017. Now corrected.

_ Fig. 1: here and elsewhere, I think the town in Sweden is called Umeå, use \aa in LATEX if you like.

Thanks. Revised. On the same line, we also corrected the spelling of Risoe to Risø.

_ Fig. 1: remove grey background from bottom panel.

Done.

_ Fig 12: the black line is the zero line - correct? Mention in the caption.

As stated in the figure caption, the figure represents a time-height cross-section of MERRA-2 vertical pressure velocity (omega) during February 2003 at the six sampling sites. Values are represented with a color code which is provided on the right of the figure. The color code is arranged to provide red for large positive (descending vertical motion) omega values and blue for large negative (ascending vertical motion) omega values; null values are indicated by the white color. The interpretation of the omega in terms of vertical motion has been added at line 445 and in the figure caption.

**References**

Charlton, A. J. and Polvani, L. M.: A new look at stratospheric sudden warmings. Part I: Climatology and modeling benchmarks, Journal of Climate, 20, 449-469, 2007.

Charlton, A. J., Polvani, L. M., Perlwitz, J., Sassi, F., Manzini, E., Shibata, K., Pawson, S., Nielsen, J. E., and Rind, D.: A new look at stratospheric sudden warmings. Part II: Evaluation of numerical model simulations, Journal of Climate, 20, 470-488, 2007.

Danielsen, E. F.: Stratospheric-tropospheric exchange based on radioactivity, ozone and potential vorticity, J. Atmos. Sci., 25, 502-518, 1968.

Dee, D. P., Uppala, S. M., Simmons, A. J., Berrisford, P., Poli, P., Kobayashi, S., Andrae, U., Balmaseda, M. A., Balsamo, G., Bauer, P., Bechtold, P., Beljaars, A. C. M., van de Berg, L., Bidlot, J., Bormann, N., Delsol, C., Dragani, R., Fuentes, M., Geer, A. J., Haimberger, L., Healy, S. B., Hersbach, H., Hólm, E. V., Isaksen, L., Kållberg, P., Köhler, M., Matricardi, M., McNally, A. P., Monge-Sanz, B. M., Morcrette, J.-J., Park, B.-K., Peubey, C., de Rosnay, P., Tavolato, C., Thépaut, J.-N., and Vitart, F.: The ERA-Interim reanalysis: configuration and performance of the data assimilation system, Q. J. R. Meteorol. Soc., 137, 553-597, https://doi.org/10.1002/qj.828, 2011.

Holton, J. R., Haynes, P., McIntyre, M. E., Douglass, A. R., Rood, R. B., and Pfister, L.: Stratosphere-troposphere exchange, Rev. Geophys., 33, 403-439, 1995.

Manney, G. L., Livesey, N. J., Santee, M. L., Froidevaux, L., Lambert, A., Lawrence, Z. D., Milln, L. F., Neu, J. L., Read, W. G., Schwartz, M. J., and Fuller, R. A.: Record-Low Arctic Stratospheric Ozone in 2020: MLS Observations of Chemical Processes and Comparisons With Previous Extreme Winters, Geophys. Res. Lett., 47, e2020GL089063, https://doi.org/10.1029/2020GL089063, 2020.

Sofieva, V., Kalakoski, N., Verronen, P., Päivärinta, S.-M., Kyrölä, E., Backman, L., and Tamminen, J.: Polar-night O3, NO2 and NO3 distributions during sudden stratospheric warmings in 2003-2008 as seen by GOMOS/Envisat, Atmos. Chem. Phys., 12, 1051-1066, 2012.

Tao, M., Konopka, P., Ploeger, F., Grooß, J.-U., Müller, R., Volk, C., Walker, K., and Riese, M.: Impact of the 2009 major stratospheric sudden warming on the composition of the stratosphere, Atmos. Chem. Phys., pp. 8695-8715, https://doi.org/10.5194/acp-15-8695-2015, 2015.

---

## Referee Report (RR1)

**Second review of "Observation and modelling of high-7Be "**

BY E. BRATTICH ET AL.

**General remarks**

In my first review I stated that the case study of high $^7$Be events observed in Northern Europe in early 2003 combined with a model simulation constitutes a good scientific study of interest to the readership of ACP. This is still true. The authors have invested considerable work in improving their manuscript in response to the comments by both reviewers.

In summary, I think this is an interesting paper the revisions have certainly improved it. I have a few remaining comments (see below) that I recommend considering when providing a final version of the paper. I suggest that the paper should now be accepted subject to technical corrections.

**Remaining detailed comments**

- Title: the title was changed in response to my comment, but I think "surface high-$^7$Be" is still not good. Perhaps "high surface $^7$Be..." or "high-$^7$Be events at the surface" or similar.

- l 55: "followed by gradual movement into the ground-level" is not really clear; do you mean the boundary layer here?

- l 86: "temporal variability of the Arctic vortex includes the SSW": includes is not the best word: perhaps 'SSW, a major mode of the temporal variability of the Arctic vortex', or similar.

- l 233: The paper states "Computation used the vertical velocity field contained in the meteorological input file" – it is still not clear which vertical velocity field you are using. I guess $\omega = \dot{p}$, or are you using a vertical velocity field in units of length over time? Please clarify.

- l 241-243: These lines do not provide a lot of discussion on downward transport and the use of backward trajectories. I still think that consideration (e.g. plotting) of time/altitude cross sections for the backward trajectories could be helpful to the arguments put forward in the paper.

- l. 441-446: These lines in the new manuscript do not really talk about PV; regarding your comment; note that for adiabatic conditions not only PV is conserved (neglecting friction) but also potential temperature (which might have implications for downward transport). You do not have to change your paper necessarily based on this comment.

- line 445: this line does not contain a discussion of omega in contrast to what is stated in the reply.

- l. 631: check authors list – does not seem to be correct.

- l. 777: check the year of the reference; 2006?

- Figure 12: The caption is extended but the meaning of the black lines in the panels of Fig. 12 is still not explained in the caption.

- Fig. 13: the omega values in the boundary are perhaps not most relevant for the downward transport from the stratosphere. Have you considered other levels as well? For example, there is a large 'red' area over Greenland (but not over the ocean surrounding Greenland); would you expect to find strong $^7$Be enhancements in Greenland but not so much over the ocean?

---

## Author Response (AR2)

**Reply to the reviewers' comments on "Observation and modeling of high-⁷Be events in Northern Europe associated with the instability of the Arctic polar vortex in early 2003" by Erika Brattich et al.**

Manuscript Ref: acp-2020-1121

We thank the reviewers for their useful comments. Below are the reviewers' comments in italic followed by our replies in blue text.

*Anonymous Referee #2*

*The manuscript has been improved by the authors and can be recommended for publication subject to a minor revision. The authors have addressed most of the previous comments and are now requested to address only one remaining, namely that about the production model.*

*The authors did not properly reply to the previous concern regarding the use of the obsolete production model by Lal & Peters (1967, LP1967) nor regarding the validity of the application of the results of 1958 to the epoch of 2003. The new text in lines 193-203 is not correct and should be rewritten. Presently it sounds as if the LP1967 model was more accurate than all the subsequent models, that is, of course, not correct. References to Koch et al. (2001) and O'Brien et al. (1991) are irrelevant here as they do not cover recent full models. This reviewer thanks the authors for pointing to the recent work of Golubenko et al. (2021) that doesn't compete with the present work but excellently shows the validity of the modern models (particularly that by Poluianov et al. 2016) as no bias between the measured data and modelled ones was found for different locations. Moreover, previous works of Usoskin et al. (2009) or Heikkila et al. demonstrated quantitative agreements between measured and modelled 7Be activities. Thus, the recent models do not produce any notable bias to the data. contrary to what the authors claimed. Thus, the authors did not properly address earlier questions. On the other hand, this reviewer agrees that precise production modelling is not crucial for this work. Thus, the reviewer suggests the following change, which would lead to a correct and acceptable text:*

*1) The new addition (lines 193-203) should be removed.*

*2) A statement ought to be added that the LP1967 model embedded into the climate model was used. While there are more accurate modern models (e.g., Masarik & Beer, 1999; Webber et al., 2007; Usoskin & Kovaltsov, 2008; Poluianov et al., 2016) the rougher one by LP1967 is sufficient for the present work where mostly atmospheric transport features are studied. Such formulation would be acceptable.*

*3) It should be clearly stated that, while the LP1967 model is used here for the time-variability studies, it is not applicable for quantitative studies of the 7Be activities.*

We thank the reviewer for his/her comments. We have followed the suggestions and revised the text (section 3.1, second paragraph).

*Anonymous Referee #3*

*General remarks In my first review I stated that the case study of high 7Be events observed in Northern Europe in early 2003 combined with a model simulation constitutes a good scientific study of interest to the readership of ACP. This is still true. The authors have invested considerable work in improving their manuscript in response to the comments by both reviewers. In summary, I think this is an interesting paper the revisions have certainly improved it. I have a few remaining comments (see below) that I recommend considering when providing a final version of the paper. I suggest that the paper should now be accepted subject to technical corrections.*

We thank the reviewer for his/her comments. Below are our point-by-point replies to each specific comment raised by the reviewer.

*Remaining detailed comments*

*ˆ  Title: the title was changed in response to my comment, but I think "surface high-7Be" is still not good. Perhaps "high surface 7Be. . . " or "high-7Be events at the surface" or similar.*

We have modified the title to "Observation and modeling of high-$^7$Be concentration events at the surface in Northern Europe associated with the instability of the Arctic polar vortex in early 2003" as suggested.

*ˆ  l 55: "followed by gradual movement into the ground-level" is not really clear; do you mean the boundary layer here?*

We have modified the sentence to "…. the transfer of stratospheric air into the upper troposphere was at its maximum in March followed by descending to the ground-level during late spring and early summer".

*☐ l 86: "temporal variability of the Arctic vortex includes the SSW": includes is not the best word: perhaps 'SSW, a major mode of the temporal variability of the Arctic vortex', or similar.*

We have modified the sentence as suggested.

*ˆ  l 233: The paper states "Computation used the vertical velocity field contained in the meteorological input file" – it is still not clear which vertical velocity field you are using. I guess ω = ṗ, or are you using a vertical velocity field in units of length over time? Please clarify.*

We have revised the sentence to "Computation used the vertical velocity (m/s) field contained in the meteorological input file"..

*^ l 241-243: These lines do not provide a lot of discussion on downward transport and the use of backward trajectories. I still think that consideration (e.g. plotting) of time/altitude cross sections for the backward trajectories could be helpful to the arguments put forward in the paper.*

Our analysis of back-trajectories arriving at the sampling sites during the two periods of low and high [7]Be values confirms the presence of two distinct circulation patterns, namely, a westerly provenance in the first period as opposed to the clockwise circulation from atmospheric upper levels (as shown from the bottom panels which present the trajectory average altitude plot, as added in the revised caption) during the second period. While this analysis alone cannot confirm the link between SSW and the [7]Be increase recorded at the sampling site, we believe that when used together with the model simulations it complements and supports our major findings with an independent method.

*^ l. 441-446: These lines in the new manuscript do not really talk about PV; regarding your comment; note that for adiabatic conditions not only PV is conserved (neglecting friction) but also potential temperature (which might have implications for downward transport). You do not have to change your paper necessarily based on this comment.*

We thank the reviewer for this comment. No changes have been made to these texts.

*^ line 445: this line does not contain a discussion of omega in contrast to what is stated in the reply.*

We have added more details on the omega values in the revised version: ",,,,, especially for the northernmost sites (Figure 12) where omega is largely positive with near-surface values up to 0.3-0.4 Pa s$^{-1}$ around 18-19 February".

*^ l. 631: check authors list – does not seem to be correct.*

The author list has been revised.

*^ l. 777: check the year of the reference; 2006?*

The year of the reference was added.

*^ Figure 12: The caption is extended but the meaning of the black lines in the panels of Fig. 12 is still not explained in the caption.*

Now the caption of Figure 12 states that the black lines are contours of omega values.

*^ Fig. 13: the omega values in the boundary are perhaps not most relevant for the downward transport from the stratosphere. Have you considered other levels as well? For example, there is a large 'red' area over Greenland (but not over the ocean surrounding Greenland); would you expect to find strong 7Be enhancements in Greenland but not so much over the ocean?*

Thanks for pointing this out. Indeed, the omega values in the boundary layer are not directly related to the downward transport from the stratosphere. Rather, positive vertical pressure velocity as seen over both Fennoscandia and Greenland (Figure 13) during 18-25 February 2003 indicates descending motions that can facilitate the transport of stratospherically influenced air, if present, to the groundlevel. This explains why Fennoscandia saw increased stratospheric influence on surface $^7$Be concentrations during this period (Figure 14), but Greenland did not see much. However, on monthly average, Greenland is a region with significant stratospheric influences in February 2003 (middle right panel, Figure 3). This discussion has been added to Lines 443-448.